# A sand fly salivary protein acts as a neutrophil chemoattractant

Anderson B. Guimaraes-Costa [1,2], John P. Shannon [3], Ingrid Waclawiak[2], Jullyanna Oliveira[2], Claudio Meneses[1], Waldione de Castro[1], Xi Wen [4], Joseph Brzostowski[5], Tiago D. Serafim [1], John F. Andersen [6], Heather D. Hickman [3], Shaden Kamhawi [1], Jesus G. Valenzuela [1✉] & Fabiano Oliveira [1✉]

Apart from bacterial formyl peptides or viral chemokine mimicry, a non-vertebrate or insect protein that directly attracts mammalian innate cells such as neutrophils has not been molecularly characterized. Here, we show that members of sand fly yellow salivary proteins induce in vitro chemotaxis of mouse, canine and human neutrophils in transwell migration or EZ-TAXIScan assays. We demonstrate murine neutrophil recruitment in vivo using flow cytometry and two-photon intravital microscopy in Lysozyme-M-eGFP transgenic mice. We establish that the structure of this ~ 45 kDa neutrophil chemotactic protein does not resemble that of known chemokines. This chemoattractant acts through a G-protein-coupled receptor and is dependent on calcium influx. Of significance, this chemoattractant protein enhances lesion pathology ($P < 0.0001$) and increases parasite burden ($P < 0.001$) in mice upon co-injection with *Leishmania* parasites, underlining the impact of the sand fly salivary yellow proteins on disease outcome. These findings show that some arthropod vector-derived factors, such as this chemotactic salivary protein, activate rather than inhibit the host innate immune response, and that pathogens take advantage of these inflammatory responses to establish in the host.

[1] Vector Molecular Biology Section, Laboratory of Malaria and Vector Research, National Institute of Allergy and Infectious Diseases, National Institutes of Health, Rockville, MD, USA. [2] Laboratório de Imunobiologia das Leishmanioses, Departamento de Imunologia, Universidade Federal do Rio de Janeiro, Rio de Janeiro, RJ, Brasil. [3] Viral Immunity and Pathogenesis Unit, Laboratory of Clinical Immunology and Microbiology, National Institute of Allergy and Infectious Diseases, National Institutes of Health, Bethesda, MD, USA. [4] Chemotaxis Section, Laboratory of Immunogenetics, National Institute of Allergy and Infectious Diseases, NIH, Rockville, MD, USA. [5] Twinbrook Imaging Facility, Laboratory of Immunogenetics, National Institute of Allergy and Infectious Diseases, NIH, Rockville, MD, USA. [6] Vector Biology Section, Laboratory of Malaria and Vector Research, National Institute of Allergy and Infectious Diseases, National Institutes of Health, Rockville, MD, USA. ✉email: jvalenzuela@niaid.nih.gov; loliveira@niaid.nih.gov

Chemokines are vertebrate-derived cytokines that attract immune cells and are members of the CC or CXC family of proteins[1,2]. These proteins have been identified in most vertebrates including zebra fish and mammals[3]. In insects, chemoattractant molecules are a family of proteins that attract insect cells, including hemocytes, and possess a unique structure and sequence that does not resemble vertebrate chemokines[4,5]. Chemokine-binding proteins have been described in tick saliva;[6] however, apart from bacterial formylated peptides and chemokine viral mimicry[7], the presence of a non-vertebrate or insect protein that can directly attract a mammalian innate cell, such as a neutrophil, has not been reported.

Insect saliva is composed of a myriad of pharmacologically active components including proteins that prevent blood clotting, vasoconstriction, and platelet aggregation needed for the insect to obtain a successful blood meal. Other activities from insect saliva have also been reported including the ability to attract white blood cells indirectly by the activation of chemokines[8].

*Leishmania* parasites are delivered into host skin by the bite of an infectious sand fly. The earliest innate cells recruited to the bite site are neutrophils, which are thought to be critical for *Leishmania* establishment in the skin of infected animals[9,10].

Together with the parasite, sand flies deliver vector-derived factors into the skin that include salivary proteins[11] and insect gut microbiota[12]. Bites of uninfected and *Leishmania*-infected sand flies recruit neutrophils in the skin of rodents in an intense and sustained fashion[10]. We have recently shown that bacteria egested from sand flies activate the inflammasome, inducing IL-1b production and consequently neutrophil recruitment[12].

As saliva is a major component of the infectious inoculum, we hypothesized that in addition to egested gut microbiota, a sand fly salivary molecule may directly participate in neutrophil recruitment. Here, we have uncovered an additional function for the yellow family of sand fly salivary proteins that directly induces chemotaxis in neutrophils, though its sequence and structure do not resemble vertebrate chemokines. We also demonstrate that *Leishmania* parasites take advantage of this insect chemoattractant protein family to establish and enhance infection in the host.

## Results

**Sand fly saliva recruits neutrophils**. A single uninfected *Phlebotomus duboscqi* sand fly was allowed to feed on the ear pinna of a wild-type C57BL/6 mouse. Two hours post sand fly bite, we observed recruitment of a significant number of CD11b+Ly6G +Ly6C$^{int}$ neutrophils to the skin (Fig. 1a and Supplementary Fig. 1) confirming previous data[13]. Neutrophil numbers peaked 6 h post bite at $\sim 7 \times 10^4$ cells/ear. By 24 h post bite, neutrophil numbers had returned to basal levels (Fig. 1a). To determine if sand fly saliva components could directly recruit neutrophils, we examined neutrophil chemotaxis induced by salivary gland homogenate (SGH) in a transwell migration assay (Fig. 1b–d). Isolated murine, human, or canine neutrophils migrated in a dose-dependent manner in the presence of sand fly SGH from two sand fly vectors, *P. duboscqi* or *Lutzomyia longipalpis* (Fig. 1b–d). Interestingly, murine neutrophils were less responsive to *L. longipalpis* SGH, and we only observed migration when we increased the concentration of SGH to ten pairs of glands (equivalent of $\sim 10\,\mu g$ of total salivary proteins) (Fig. 1b). In contrast, human neutrophils (Fig. 1c) and canine neutrophils (Fig. 1d) responded efficiently to *L. longipalpis* SGH and to a lesser extent to *P. duboscqi* SGH (Fig. 1b, c). Adding ten pairs of *P. duboscqi* SGH to both the top and bottom transwell chambers disrupted directional neutrophil migration compared to the control (Fig. 1e).

To validate and visualize neutrophil migration in response to *P. duboscqi* SGH, we performed an EZ-TAXIScan assay with human neutrophils attached to a fibronectin coated surface[14] (Fig. 1f–h). When using ten pairs of SGH (10 μg of salivary protein), human neutrophils became polarized and migrated towards *P. duboscqi* SGH (Supplementary Movie 1) at a speed and rate similar to N-Formylmethionyl-leucyl-phenylalanine (fMLP), the positive control (Supplementary Movie 2). Of note, despite migrating at a similar speed, we observed that neutrophils took longer to sense the SGH gradient, possibly due to the complex mixture of proteins (around 35 distinct proteins) present in sand fly saliva. As a negative control, neutrophils exposed to RPMI/0.1% BSA alone (Supplementary Movie 3) exhibited random migration (Fig. 1f–h). Altogether, these results strongly suggest that sand fly SGH can induce directional, dose-dependent neutrophil migration.

**Identification of the chemoattractant factor in sand fly saliva**. To characterize and identify the factor responsible for neutrophil migration in sand fly saliva, we first treated *P. duboscqi* SGH with proteinase K and performed a transwell migration assay using murine neutrophils (Fig. 2a). *P. duboscqi* SGH treated with proteinase K resulted in loss of neutrophil chemotaxis, strongly suggesting that a protein in *P. duboscqi* SGH was at least partially responsible for neutrophil chemotaxis.

To test for the possibility that lipids in *P. duboscqi* SGH may possess chemotactic activity, we treated *P. duboscqi* SGH with ethanol to precipitate protein moieties while retaining any lipids in the supernatant. The factor(s) eliciting to neutrophil migration were retained in the pellet and were not observed in response to the lipid-enriched supernatant (Fig. 2b), indicating that lipids present in SGH were not mediating neutrophil chemotaxis. Finally, heating the active SGH protein pellet to 70 °C for 30 min resulted in loss of neutrophil chemotaxis (Fig. 2b). Together, these data suggest that a protein in sand fly saliva promotes neutrophil chemotaxis and that the tertiary structure of this protein is important for chemotactic activity.

To identify the protein responsible for neutrophil chemotaxis in sand fly saliva, we chose to test the most abundant proteins in saliva of *P. duboscqi* and *L. longipalpis* sand flies: PduM10, PduM35, LJM11, LJM111, and LJM17 from the yellow family of proteins, PduM34 from the silk related family of proteins, and LJL143 from the Lufaxin family of proteins. We expressed these sand fly salivary proteins in HEK cells, purified them as endotoxin-free recombinant proteins (Supplementary Fig. 2), and tested the purified proteins for chemotactic activity. We tested *P. duboscqi* recombinant proteins in a transwell migration assay using murine neutrophils, while *L. longipalpis* recombinant proteins were tested in human neutrophils to maximize their migration potential according to data in Fig. 1b–d. Only recombinant proteins corresponding to members of the yellow family of proteins from the saliva of both *P. duboscqi* (rPduM10, rPduM35) and *L. longipalpis* (rLJM11, and rLJM17) exhibited chemotactic activity in the EZ-TAXIScan assays. We did not observe a chemotactic activity for the recombinant salivary proteins rLJM111, rLJL143, and rPduM34 (Supplementary Fig. 3). Compared to the control, recombinant yellow salivary proteins rPduM10 and rPduM35 from *P. duboscqi* elicited murine neutrophil migration with the latter displaying a dose-dependent effect (Fig. 2c). Additionally, neutrophil migration was significantly increased when the two proteins (rPduM10 and rPduM35) were added together (Fig. 2d). *L. longipalpis* recombinant yellow salivary proteins rLJM17 and rLJM11 also elicited human neutrophil migration in a dose-dependent manner (Fig. 2e), and neutrophil migration was significantly increased

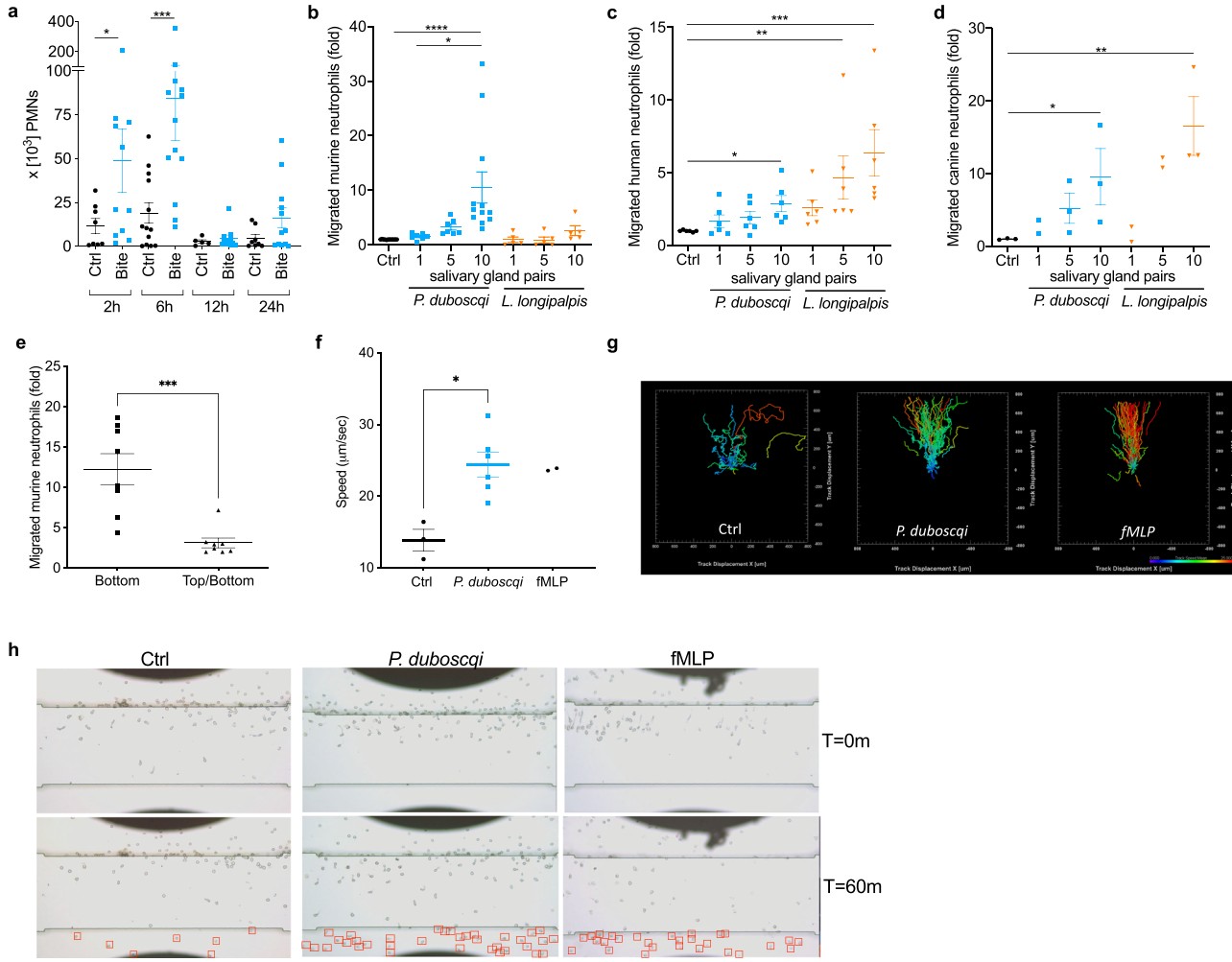

**Fig. 1 *P. duboscqi* and *L. longipalpis* salivary glands homogenate contains neutrophil chemoattractants. a** Flow cytometry analysis of in vivo neutrophils recruitment to C57BL6 mice ears. Mice ears were exposed to a single bite of *P. duboscqi* and neutrophil (Ly6G$^{high}$/CD11b$^+$/Ly6C$^{low/-}$) recruitment was analyzed after different time points. **b**, **d** Neutrophils were added to the top chamber of a transwell plate and different amounts of the sand fly salivary gland homogenate were added to the bottom chamber. After 3 h, cells that migrated to the bottom chamber were quantified by DNA content. **e** Neutrophils were added to the top chamber of a transwell plate and ten pairs of *P. duboscqi* salivary glands were added to the bottom chamber or to both (top and bottom) chambers and run as in **b**–**d**. **b**, **e** Murine neutrophils were purified by magnetic negative selection from C57BL/6 bone marrow cells. **c** Human and **d** canine neutrophils were purified from peripheral blood of healthy donors by density gradient. **f**–**h** Validation of sand fly salivary gland homogenate driven human neutrophil recruitment using EZ-TAXIScan assay. N-Formylmethionyl-leucyl-phenylalanine (fMLP) was used as a positive control. **f**, **g** Chemotaxis outcomes were analyzed by Imaris. **f** Chemotaxis mean speed was computed as the total path length divided by time. **g** Each track is plotted from the central point and shows XY displacement. **h** Still images show neutrophils at two different time points ($T = 0$ m and $T = 60$ m). Red squares show migrated neutrophils at the bottom of the chamber. Data are presented as means ± SEM. **a**, **e**–**g** Cumulative results of two independent experiments. **b**–**d** Cumulative results of 3 independent experiments. **a** $n = 8, 11, 13, 13, 5, 11, 8, 12$; *$P = 0.0343$; ***$P = 0.0003$. **b** $n = 12, 7, 7, 12, 5, 5, 5$; *$P = 0.017$; ****$P < 0.0001$. **c** $n = 6$; *$P = 0.044$; **$P = 0.0026$; ***$P = 0.0001$. **d** $n = 3, 2, 3, 3, 2, 2, 3$; *$P = 0.0389$, *$P = 0.0312$; **$P = 0.0022$. **e** $n = 8$; ***$P = 0.0006$. **f** $n = 3, 6, 2$, *$P = 0.0238$; $P$ values calculated by two-tailed Mann–Whitney test (**a**, **e**, **f**) or one-way ANOVA (**b**–**d**).

when these two proteins were added together (Fig. 2f). Amino acid identity between *P. duboscqi* rPduM10 and rPduM35, and *L. longipalpis* rLJM17 and rLJM11 yellow proteins range from 43.6 to 48.8%. Alignment shows several conserved areas of amino acids across these proteins (Supplementary Fig. 4a) that were also maintained across salivary yellow proteins from other sand flies[15] (Supplementary Fig. 4b), suggesting that yellow salivary proteins from other sand flies may also exhibit neutrophil chemoattractant activity. We confirmed the transwell data using the EZ-TAXIScan assay (Fig. 2g–i). Human neutrophils promptly polarized and migrated towards a 100 nM gradient of rPduM10 (Supplementary Movie 4). Moreover, cells migrated towards rPduM10 with a similar speed (Fig. 2g) and directionality (Fig. 2h) as the positive control fMLP (Supplementary Movie 5). rPduM10-induced

neutrophil migration was distinct from the negative control group (Supplementary Movie 6) which resulted in lower speed and no directionality (Fig. 2h). The recombinant protein rPduM34, belonging to the silk family of proteins, did not induce a directional migration of neutrophils (Supplementary Fig. 3), suggesting that the two members of the sand fly yellow-related family of proteins in *P. duboscqi* sand fly saliva are responsible for induction of neutrophil migration. We then tested if the migration towards yellow proteins was dependent on neutrophil G-protein-coupled receptors (GPCR), phosphatidyli-nositol 3-kinases (PI3K) activity, or calcium influx. When neutrophils were pretreated with either pertussis toxin (PTx) (blocking GPCR-mediated migration) or BAPTA-AM (which chelates calcium), their migration toward rPduM10/35 (Fig. 2j)

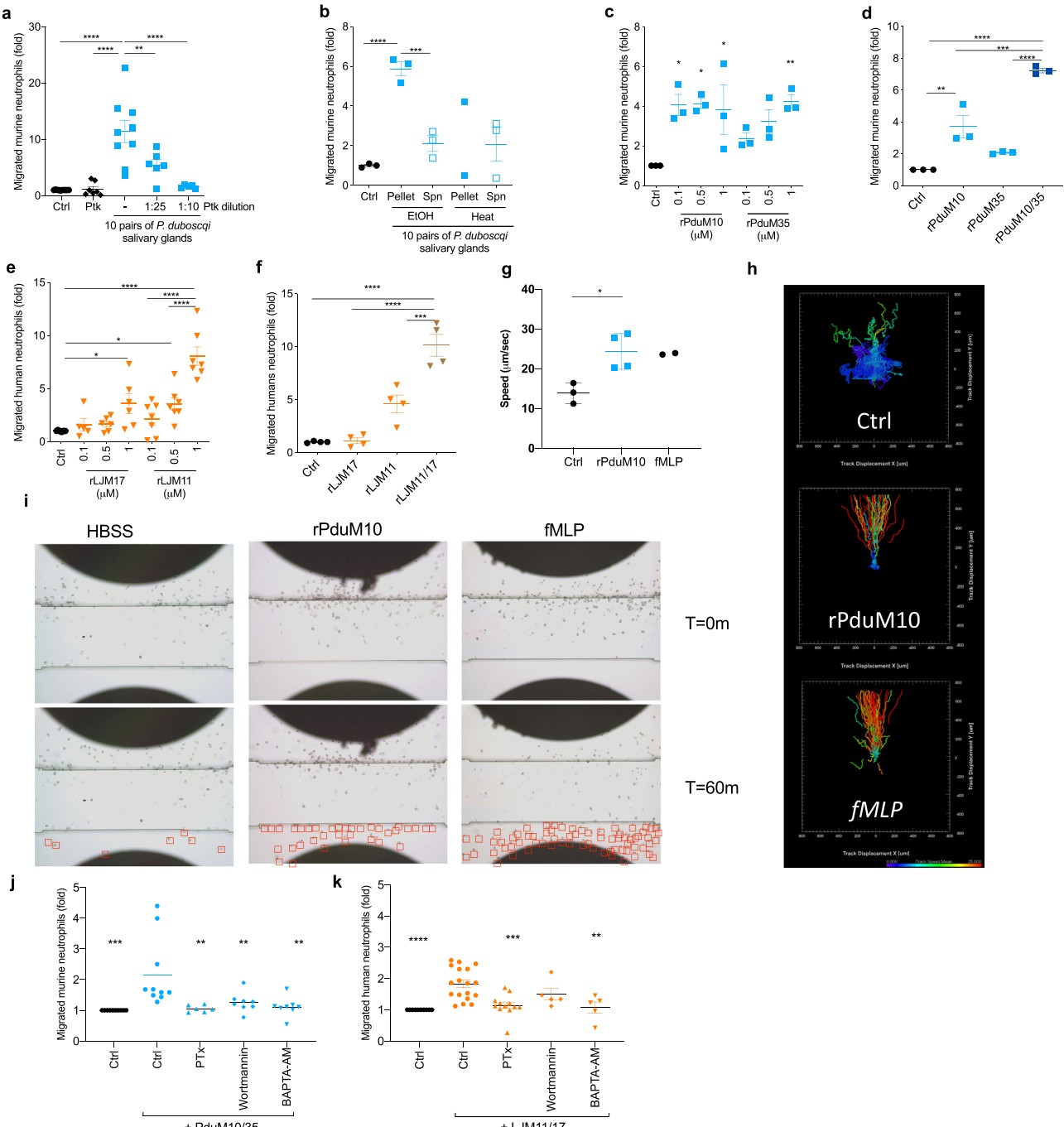

**Fig. 2 Sand fly salivary yellow proteins are the neutrophil chemoattractants in sand fly saliva.** Neutrophils from C57BL6 bone marrow (**a–d**, **j**) or from human periheral blood (**e**, **f**, **k**) were added to a transwell plate top chamber and *P. duboscqi* salivary gland homogenate (SGH) or recombinant proteins were added to the bottom chamber. After 3 h, cells in the bottom chamber were quantified. *P. duboscqi* SGH was treated with Proteinase K (Ptk) (**a**), heated for 30 min at 70 °C (**b**) or treated with 90% of ethanol (EtOH) (**b**). Increasing amounts of rPduM10 or rPduM35 (**c**); rPduM10 (0.5 μM), rPduM35 (0.5 μM) or rPduM10 + rPduM35 (0.25 μM of each protein) (**d**), increasing amounts of rLJM17 or rLJM11 (**e**), or rLJM17 (0.5 μM), rLJM11 (0.5 μM) or rLJM17 + rLJM11 (0.25 μM of each protein) (**f**). **g–i** Validation of yellow proteins as neutrophil chemoattractants by EZ-TAXIScan assay. N-Formylmethionyl-leucyl-phenylalanine (fMLP) was used as a positive control, and Hanks' Balanced Salt Solution (HBSS) as baseline control. **g**, **h** Chemotaxis parameters were analyzed by Imaris. **g** Chemotaxis mean speed was computed as total path length divided by time. **h** Each track is plotted from the central point showing XY displacement. **i** Still images of neutrophils at two different time points. Red squares show migrated neutrophils. Neutrophils were pretreated with pertussis toxin (PTx; 2 μg/mL), wortmannin (1 μM) or BAPTA-AM (10 μM) and their migration towards a combination of rPduM10 + rPduM35 (**j**) or rLJM11 + rLJM17 (**k**) (0.25 μM each) was assayed. Cumulative results of at least three (**a–f**, **j**, **k**) or two (**g–i**) independent experiments are presented as means ± SEM; **a** $n = 10, 7, 9, 6, 5$; ****$P = 0.0043$, *****$P < 0.0001$. **b** $n = 3, 3, 3, 2, 3$; *****$P = 0.0003$, ******$P < 0.0001$. **c** $n = 3$; *$P = 0.0123$, *$P = 0.0104$, *$P = 0.0208$, **$P = 0.0081$. **d** $n = 3$; **$P = 0.0042$, ***$P = 0.0007$, ****$P < 0.0001$. **e** $n = 8, 5, 6, 5, 7, 7, 7$; *$P = 0.0446$, **$P = 0.0387$, ****$P < 0.0001$. **f** $n = 4$; ***$P = 0.0003$, ****$P < 0.0001$. **g** $n = 3, 4, 2$; *$P = 0.0162$. **j** $n = 10, 10, 6, 8, 8$; **$P = 0.0032$, **$P = 0.0093$, **$P = 0.0021$ ***$P = 0.0003$. **k** $n = 10, 18, 12, 5, 5$; **$P = 0.0018$ ***$P = 0.0001$, ****$P < 0.0001$. $P$ calculated by one-way ANOVA test, except (**g**) two-tailed unpaired $t$-test.

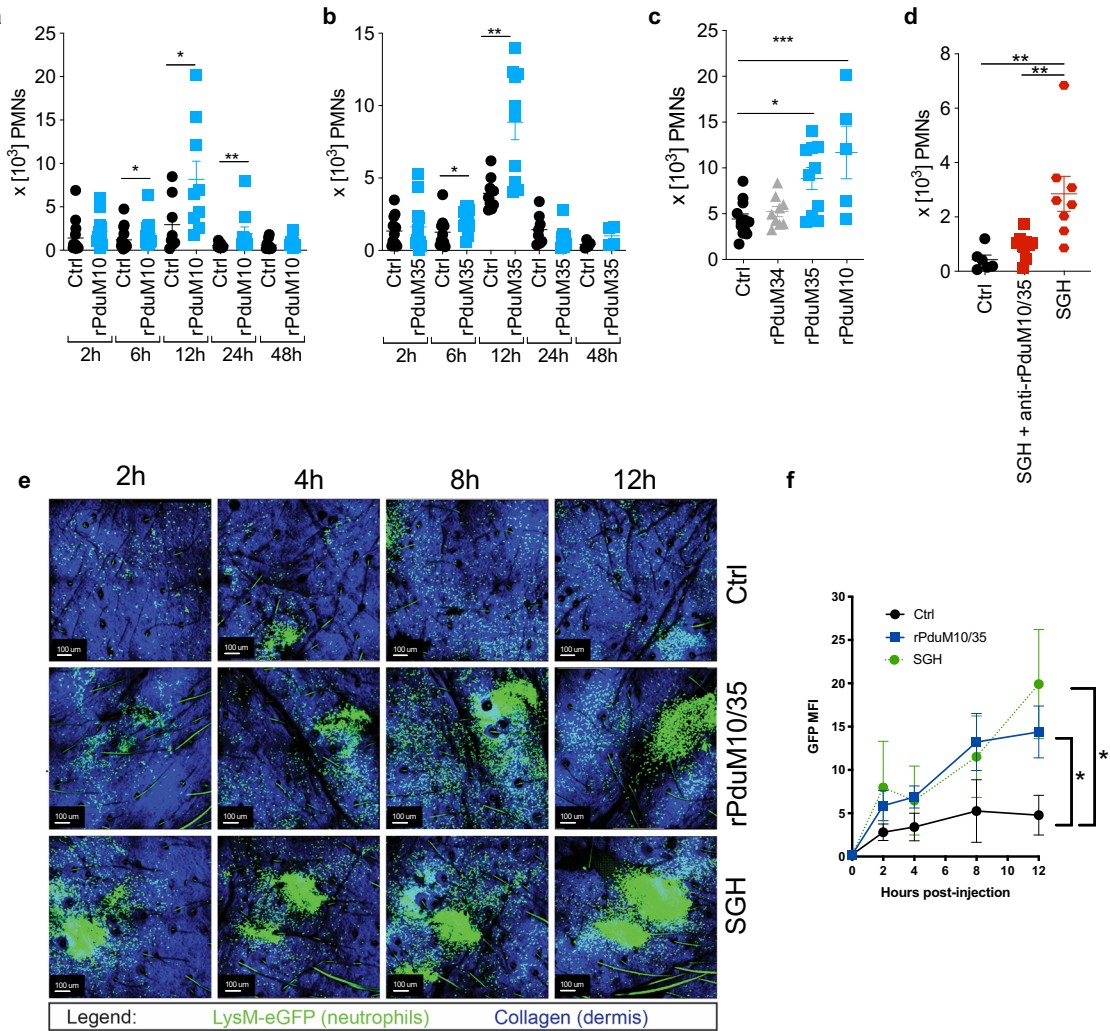

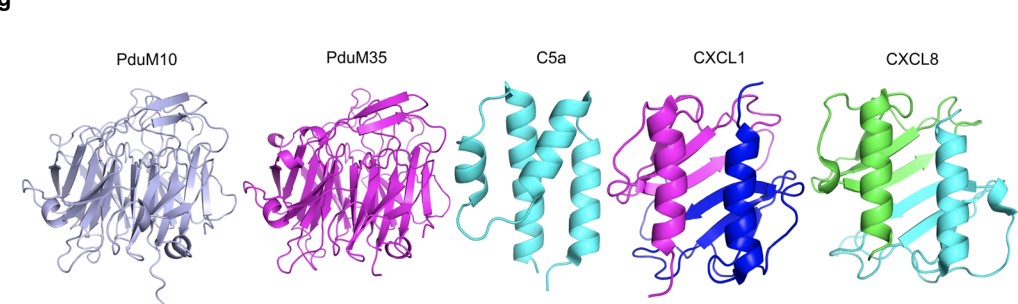

and rLJM11/17 (Fig. 2k) was abrogated in the transwell assay. Interestingly, pretreatment of neutrophils with wortmannin, a pan-PI3K inhibitor, abolished migration to rPduM10/35 (Fig. 2j), but some migration towards rLJM11/17 persisted (Fig. 2k).

**Sand fly salivary yellow proteins recruit neutrophils in vivo.** After demonstrating that recombinant sand fly yellow salivary proteins induced neutrophil migration in vitro, we next tested whether neutrophil recruitment also occurred in vivo. Flow cytometric analysis of single-cell suspensions of mouse ear pinnae injected with 0.25 μM of either rPduM10 or rPduM35 recruited a significantly higher number of neutrophils in vivo compared to

controls (Fig. 3a, b). For both proteins, enhanced recruitment of neutrophils over control was evident in the tissue at 12 h post injection, returning to baseline by 48 h (Fig. 3a, b). Inoculation of the same quantity of rPduM34, a member of the silk family of proteins that did not elicit in vitro neutrophil migration, did not recruit neutrophils in vivo (Fig. 3c). Additionally, we tested a combination of 0.25 μM of *L. longipalpis* yellow proteins rLJM11 + rLJM17 in vivo and observed a statistically significant neutrophil recruitment 12 h post injection (Supplementary Fig. 5), corroborating our in vitro data on the chemoattractant role of the yellow proteins from both *L. longipalpis* and *P. duboscqi* in vivo. Of note, the amount of protein in a pair of salivary glands varies by sand fly species[16,17], and a sand fly

**Fig. 3 Yellow-related salivary proteins act as chemoattractants for neutrophils in vivo. a–d** Flow cytometric analysis of neutrophil (Ly6G[high]/CD11b[+]/Ly6C[low/−]) recruitment in C57BL/6 mouse ears. Mice ears were injected with HBSS as negative control (Ctrl) or with **a** rPduM10 (0.25 μM), **b** rPduM35 (0.25 μM), **c** rPduM34, rPduM35 or rPduM10 (0.25 μM). Neutrophil recruitment was analyzed at different time points as shown in (**a**, **b**) or at 12 h for (**c**). **d** Mice ears were injected with salivary gland homogenate (SGH) treated or not treated with antibodies against PduM10 or PduM35. Neutrophil recruitment was analyzed after 12 h. **e**, **f** Intravital multiphoton imaging of neutrophil migration in eGFP-LysM C57BL/6 mouse ears. Ears were injected with HBSS as a negative control, rPduM10 + rPduM35 (0.25 μM of each) or SGH (0.5 pair). Neutrophil migration was recorded overtime. **e** Still images from a representative experiment show neutrophil migration (green) at different time points. **f** Neutrophil accumulation overtime is shown as GFP MFI. **g** Ribbon depictions of predicted three dimensional structures of rPduM10 (light blue) and rPduM35 (magenta) obtained by modeling using I-TASSER, compared with the structures of C5a (cyan, https://doi.org/10.2210/pdb4P3A/pdb) determined by X-ray crystallography, CXCL1 (blue and magenta, https://doi.org/10.2210/pdb1MGS/pdb) determined by NMR and CXCL8 determined by X-ray crystallography (green and cyan, https://doi.org/10.2210/pdb1IL8/pdb). Cumulative results of two (**a–d**) or three (**e**, **f**) independent experiments are presented as means ± SEM; **a** n = 13, 14, 16, 18, 8, 9, 10, 10, 10, 10; *P = 0.0136, *P = 0.0360, **P < 0.0089. **b** n = 11, 13, 10, 12, 10, 10, 8, 4, 4; *P = 0.0237, **P = 0.0021. **c** n = 12, 10, 9, 5; *P = 0.0120, ***P = 0.0009. **d** n = 6, 8, 10; **P = 0.0030, **P = 0.0011. **f** n = 3, 4, 3. *P < 0.0462, **P < 0.0223, calculated by **a**, **b** two-tailed Mann–Whitney test, **c**, **d** one-way ANOVA and **f** two-way ANOVA test.

delivers 70–90% of its salivary protein content while feeding[18–20]. For *P. duboscqi* and *L. longipalpis*, a pair of salivary glands contains between 0.78 to 1 μg of protein[17,20] and 0.5 to 1 μg of protein[21,22], respectively. Further, transcriptomic analysis indicates that the relative abundance of yellow proteins in the salivary glands of *L. longipalpis* and *P. duboscqi* is 16.0%[23] and 7.5%[24], respectively. Based on the above, and from a recently reported estimate of the amount and proportion of yellow salivary proteins present in sand flies[25], we estimate there is ~70 ng of yellow proteins in *P. duboscqi* and ~120 ng of yellow proteins in *L. longipalpis* salivary glands. Therefore, the tested recombinant protein concentration of 0.25 μM in an injection volume of 10 μL (=100 ng) is within the physiological range of yellow proteins injected during a sand fly bite[25].

To examine the contribution of the two yellow proteins to neutrophil recruitment in vivo by *P. duboscqi* saliva, we generated polyclonal antibodies against the recombinant proteins rPduM10 and rPduM35 (Supplementary Fig. 6). We incubated these polyclonal antibodies with whole *P. duboscqi* SGH, then injected the mixture into a mouse ear pinna (Fig. 3d). Antibodies to rPduM10 and rPduM35 abrogated neutrophil recruitment by *P. duboscqi* SGH, indicating that these two proteins are the major neutrophil-chemotactic factors present in sand fly saliva.

To understand the spatial dynamics of neutrophil recruitment in response to sand fly chemotactic proteins, we co-injected rPduM10 and rPduM35 in the mouse ear pinna and visualized neutrophils by two-photon intravital microscopy in Lysozyme-M-eGFP transgenic mice[26] with GFP+ neutrophils. After co-injection of rPduM10 and rPduM35 (0.25 μM), there was a marked recruitment of neutrophils into the ear pinna as early as 2 h post injection (Supplementary Movie 5). Neutrophil recruitment increased over time and was sustained up to 12 h post injection as compared to a weaker and transient response with PBS injection alone (Fig. 3e). A strong recruitment of neutrophils into the ear pinna was also observed with sand fly SGH (Fig. 3e and Supplementary Movie 7). To quantify recruited neutrophils in the images, we calculated the GFP mean fluorescent intensity (MFI) per ear, which demonstrated that rPduM10/rPduM35 co-injection resulted in similar GFP signals (neutrophil recruitment) to those seen after whole SGH injection (Fig. 3f). Together, these data further support our conclusion that rPduM10 and rPduM35 are the major neutrophil recruiting salivary proteins in *P. duboscqi* saliva.

In contrast to the small size of vertebrate chemokines (~ 8–11 kDa proteins), the sand fly yellow family member proteins are large proteins of ~43 kDa[27]. To further evaluate the nature of the two insect salivary proteins in the context of known neutrophil chemoattractants, we compared their structure to known vertebrate chemokines. We obtained high-confidence models of the *P. duboscqi* proteins using the structure of the yellow family member protein LJM11 from *L. longipalpis*[28] as a template. Our modeling indicates that the sand fly yellow family of proteins are monomeric and have a six-bladed β-propeller fold containing a binding site for a single biogenic amine ligand molecule at one end[15] (Fig. 3g). A similar structure was independently reported for these same yellow proteins[15]. Models of Pdub1 (equivalent to PduM10) and Pdub2 (equivalent to PduM35) were very similar with PduM10 and Pdub1 showing a root-mean-square deviation (rmsd) of 0.20 Å over 304 Cα positions and PduM35 and Pdub2 showing a rmsd value of 0.21 Å over 307 Cα positions reinforcing accuracy of the models. In comparison, the neutrophil recruiting chemokines CXCL1 or CXCL8 have a symmetrical dimeric structure containing a single six-stranded β-sheet made up of three strands from each monomer laid up against two α-helices also contributed by the two monomers (Fig. 3g). The structure of C5a, a chemotactic complement component, consists of a four α-helix bundle and also does not resemble the β-propeller structure of the yellow proteins (Fig. 3g). These data indicate that the neutrophil chemoattractant proteins, PduM10 and PduM35 bear no similarity to the structure of the vertebrate neutrophil-attracting proteins CXCL1, CXCL8, or C5a.

**Co-injection of *Leishmania* with rPduM10 and rPduM35 increases disease severity.** Sand fly salivary proteins are naturally co-delivered into the skin along with *Leishmania* parasites. To determine the effect of the two neutrophil chemoattractant proteins on *Leishmania* infection, we injected 500 *Leishmania major* metacyclics, similar to the median number of parasites reported to be delivered by a sand fly bite[29], either alone or in conjunction with 0.25 μM of recombinant rPduM10 and rPduM35. Co-injection of *Leishmania major* with *P. duboscqi* SGH, or with combined rPduM10/rPduM35 significantly increased *Leishmania* pathology and parasite load compared to injection of parasites alone or along with negative control, recombinant protein rPduM34 (Fig. 4a–c). By 3 weeks post infection, we observed a significant increase in ear thickness, indicative of a cutaneous leishmaniasis lesion (Fig. 4a). Additionally, the parasite burden was significantly enhanced at 1 week post infection in mice that received co-injection of *Leishmania* with rPduM10/35 or SGH compared to injection of parasites alone (Fig. 4b), and this enhancement was maintained up to 7 weeks post infection (Fig. 4c). Importantly, preincubation of parasites with *P. duboscqi* SGH plus antibodies to rPduM10/rPduM35 significantly reduced the parasite burden compared to mice only co-injected with parasites and *P. duboscqi* SGH, at both 1 week (Fig. 4d) and 7 weeks (Fig. 4e) post infection.

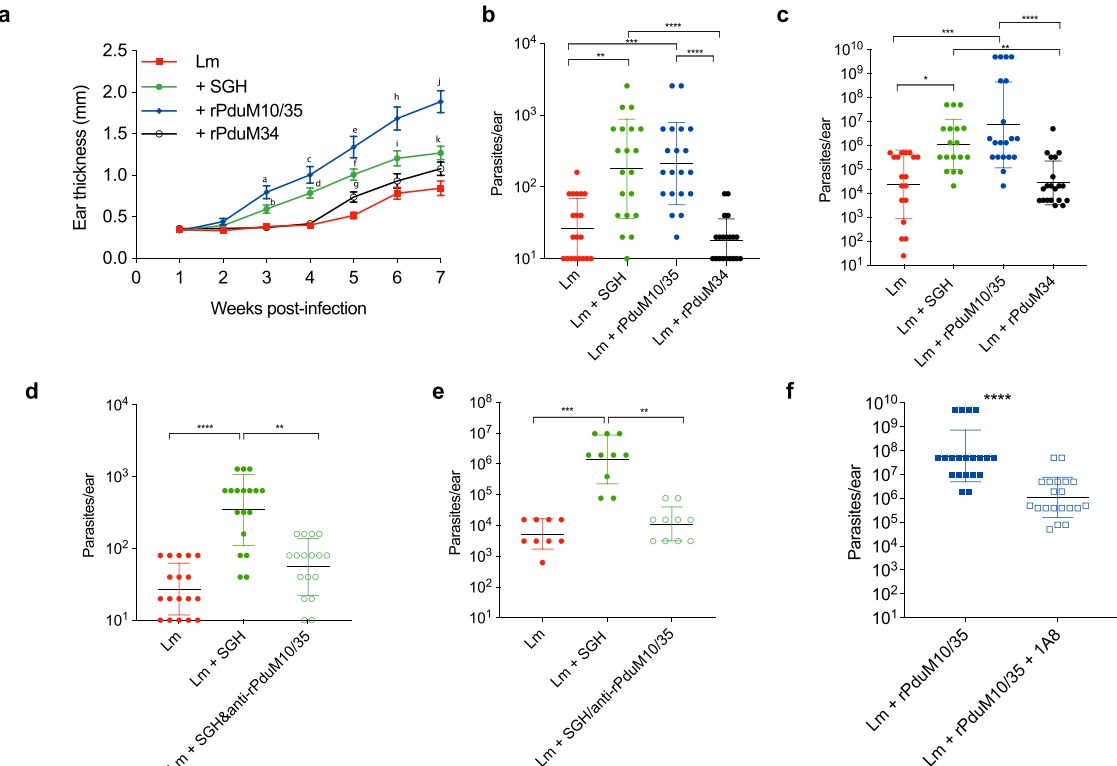

**Fig. 4 Sand fly salivary yellow proteins exacerbate *Leishmania* infection. a–c** C57BL6 mice ears were infected with *L. major* metacyclics (500 parasites) in PBS or with rPduM10 + rPduM35 (0.125 μM of each), *P. duboscqi* SGH (0.5 pair) or rPduM34 (0.25 μM). **a** Ear thickness was measured every week. Parasite load assessed by limiting dilution assay at **b** 1 or **c** 7 weeks post infection. **d**, **e** C57BL/6 mouse ears were infected with *L. major* metacyclics (500 parasites) in PBS or with 0.5 pair equivalent of *P. duboscqi* SGH premixed or not with antibodies against PduM10 or PduM35 (1:100 dilution). Parasite load assessed by limiting dilution assay at 1 (**d**) or 6 weeks (**e**) post infection. **f** Anti-Ly6G (clone 1A8) antibody was administered intraperitoneally to C57BL/6 mice 18 h before infection. C57BL6 mouse ears were infected with *L. major* metacyclics (500 parasites) in PBS or with rPduM10 + rPduM35 (0.125 μM of each). Parasite load assessed by limiting dilution assay 7 weeks post infection. Cumulative results of two independent experiments are presented as **a** means ± SEM or **b–f** geometric mean ± geometric SD; **a** $n = 20, 20, 18, 20$. a $P = 0.031$, b $P = 0.024$, c $P = 0.0002$. d $P = 0.0013$. e, f, h, j $P < 0.0001$. **g** $P = 0.0149$. i $P = 0.021$. k $P = 0.0028$. **b** $n = 20$; **$P = 0.0014$, ***$P = 0.0002$, ****$P < 0.0001$. **c** $n = 18, 18, 20, 20$; *$P = 0.162$, **$P = 0.0014$, ***$P = 0.0005$, ****$P < 0.0001$. **d** $n = 18, 18, 17$; **$P = 0.0027$, ****$P < 0.0001$. **e** $n = 9, 10, 10$; **$P = 0.0017$, ***$P = 0.0001$. **f** $n = 20$; ****$P < 0.0001$, calculated by **a** two-way ANOVA test, **b–e** one-way ANOVA or **f** two-tailed Mann–Whitney test.

We next tested whether the increase in parasite burden observed upon co-injection of parasites and recombinant salivary proteins was related to neutrophil recruitment induced by rPduM10/35 at the site of infection. It has been shown that transient neutrophil depletion during needle injection of *Leishmania* parasites or sand-fly delivered *Leishmania* leads to a diminished parasite ability to establish infection in the C57BL/6 mice[10,30]. We, therefore, depleted neutrophils prior to infection using injection of 1A8 (Supplementary Fig. 7), a monoclonal antibody against Ly6G that is standardly administered to transiently deplete neutrophils while maintaining monocytes and macrophages[31]. We observed that neutrophil depletion also reduced parasite burden in rPduM10/rPduM35-injected animals at 7 weeks post infection (Fig. 4f). These data suggest that neutrophils recruited to the tissue by rPduM10/rPduM35 upon injection contribute to enhanced parasite growth. Taken together, these data strongly suggest that rPduM10/rPduM35 significantly enhance *Leishmania* infection increasing both parasite burden and pathology through augmentation of neutrophil recruitment into the skin via their chemotactic activity.

## Discussion

Chemokines are a family of small secreted cytokines that control the migration of white blood cells to sites of infection or tissue injury[1,3]. These molecules interact with target cells through GPCR and have been classified into four subfamilies: CXC, CC, CX3C, and XC according to specific structural characteristics[3]. Here, we identify a family of mammalian cell chemoattractants from saliva of sand flies. We demonstrate that two members of the yellow family of salivary proteins from two vector species of sand flies exert directional chemotaxis in human, dog, and mouse neutrophils. These chemoattractant proteins have a molecular weight of ~43 kDa and share no sequence or structural signatures with *bona fide* chemokines[32,33] or other known chemoattractants. These sand fly salivary yellow protein chemoattractants follow a classical pathway of GPCR-activated neutrophil migration[34], that is dependent on calcium influx and, for rPduM10/35, on PI3K signaling[35,36]. Unexpectedly, the inhibition of PI3K by wortmannin was not sufficient to abrogate migration towards rLJM11/17. Further studies with PI3K-deficient neutrophils and other PI3K selective inhibitors will corroborate if rLJM11/17 chemotaxis is PI3K independent, similar to what has been observed in LPS-primed humans cells[37]. Interestingly, rLJM111, a third member of the yellow family of salivary proteins in *L. longipalpis*, did not show neutrophil-chemotactic activity in our assays. LJM111 is phylogenetically distant from the yellow salivary protein LJM17 and less distant to LJM11[23]. Furthermore, it has been demonstrated that rLJM111 has anti-inflammatory activity[38], suggesting that its biological activity is different from LJM11 and

LJM17 that have now been characterized as neutrophil chemoattractants. To our knowledge, this is the first description of an insect or arthropod molecule with chemotactic activity for mammalian immune cells. These salivary chemoattractant molecules are also structurally distinct from insect cytokines such as the hemocyte chemotactic peptide present in moths[5].

The yellow family of proteins were first characterized in Drosophila larvae and pupae as dopachrome-conversion enzymes involved in melanization[39]. A similar protein (with ~30–40% homology) with the same enzymatic activity[40] was later found in larvae and pupae of Aedes aegypti mosquitoes. Thus far, the presence of the yellow family of proteins has only been confirmed in saliva of sand flies[11].

Previously, our group demonstrated that yellow salivary proteins of L. longipalpis bind and trap biogenic amines, such as serotonin, histamine, and catecholamines[28]. More recently, it was shown that the yellow proteins from the genus Phlebotomus also have the function of biogenic amine binding proteins[25] demonstrating this biological function is present in these relevant vectors of Leishmania. We have now shown a dual function of this abundant and ubiquitous sand fly family of yellow salivary proteins as a neutrophil chemoattractant that recruits cells to the bite site. Thus far, the presence of the yellow family of proteins has only been confirmed in saliva of sand flies. In fact, this family of salivary proteins have been characterized as biomarkers for sand fly exposure on humans and dogs[11,16,41] and recently a diagnostic test was developed based on this family of sand fly salivary proteins using a recombinant protein from the sand fly P. perniciosus[42].

Neutrophils play an important part in the establishment of Leishmania parasites in the host[43,44]. For L. major, several studies have demonstrated that neutrophils promote early parasite survival by acting as trojan horses, temporarily shielding viable parasites from adverse conditions until they transition into macrophages, their permanent host cell[10]. Our data support these findings as co-inoculation of L. major parasites with the yellow salivary proteins augmented recruitment of neutrophils, increased parasite burden, and enhanced disease pathology. Importantly, we established that the chemotactic activity of the yellow salivary proteins for neutrophils is directly responsible for disease enhancement.

Yellow salivary proteins are not the only molecules in sand fly saliva to modulate the host immune response. Sand fly saliva also contains endonucleases that counteract the effect of neutrophil NETosis, protecting Leishmania parasites and worsening disease[45]. Moreover, a still unidentified sand fly salivary molecule was shown to induce neutrophil apoptosis, promoting parasite survival in these cells[46]. Recently, our group demonstrated that gut microbes egested into skin during the bite of Leishmania-infected sand flies initiate an acute inflammatory response mediated by IL-1b-driven neutrophil recruitment[12]. Earlier, Rogers et al.[47]. demonstrated that proteophosphoglycans secreted by Leishmania parasites during midgut development are egested and modulate the host immune response, primarily through enhanced recruitment of macrophages and increased macrophage arginase activity[48,49]. Identification of the chemoattractant activity of yellow salivary proteins for neutrophils furthers our understanding of the diverse molecules at play modulating the host skin environment. Of note, it is clear that there is a redundancy in innate immune cellular recruitment to the skin following an infected vector bite, potentially indicative of an evolutionary adaptation by Leishmania in sand flies to ensure parasite survival in the hostile host skin environment.

The benefit to sand flies of attracting neutrophils remains unclear and warrants further investigation. However, this phenomenon is not unique to sand flies. Neutrophil recruitment or activation of the innate immune response by insect or tick bites, or by insect- or tick-derived factors, has been shown by different studies to, counterintuitively, augment pathogen infectivity[12,50–53]. This includes work by Pingen et al.[52]. who demonstrated that bites of A. aegypti induce a rapid recruitment of neutrophils resulting in higher viremia and increased severity of disease as compared to needle inoculation of Semliki Forest virus [52]. Additionally, recent work by Uraki et al.[53] showed that the salivary protein AgBR1 from A. aegypti promotes neutrophil recruitment and that antibodies against this salivary protein decrease neutrophil recruitment and protect animals against Zika infection[53], providing further evidence that activation of innate cells by vector-derived factors is important for pathogen infectivity.

In summary, we have identified a family of neutrophil chemoattractant proteins from saliva of sand flies. Proteins with similar functionality may be present in the saliva of other blood-feeding arthropods, evolved to alter the skin innate immune response of the host. Activation of the innate immune system by vector-derived factors appears to provide a safe niche for pathogens delivered by vectors, promoting their establishment in mammalian hosts, and ultimately, manifestation of disease.

## Methods

**Animals**. C57BL/6 and Balbc 4–6 weeks old female mice were purchased from Charles Rivers (Wilmington, MA). LysM-eGFP mice were acquired through the NIAID Taconic Research Repository and bred in house. We have complied with all relevant ethical regulations for animal testing and research. All animal experimental procedures were reviewed and approved by the National Institute of Allergy and Infectious Diseases (NIAID) Animal Care and Use Committee under animal study protocol LMVR-4e. Dog blood was obtained from the Division of Veterinary Research at the National Institutes of Health.

**Human blood source**. We have complied with all relevant ethical regulations for work with human participants. Human Peripheral blood mononuclear cells from healthy subjects were collected under written informed consent from participants of the National Institutes of Health Clinical Center IRB-approved protocol number 99-CC-0168, entitled Collection and Distribution of Blood Components from Healthy Donors for In Vitro Research Use, ClinicalTrials.gov Identifier: NCT00001846.

**Parasites**. Leishmania major promastigotes (WR 2885 strain) were cultured at 26 °C in Schneider's medium supplemented with 10% heat-inactivated fetal bovine serum (FBS), 100 U/mL penicillin, 100 μl/mL streptomycin and 2 mM l-glutamine. Infective-stage metacyclic promastigotes of L. major were isolated from stationary cultures (5 days old) by negative selection using peanut agglutinin (PNA; Vector Laboratories Inc).

**Sand flies and salivary glands homogenate (SGH) preparation**. P. duboscqi (Mali strain) and L. longipalpis (Jacobina strain) were reared at the Laboratory of Malaria and Vector Research, NIAID, NIH. Salivary glands were dissected from 5 to 7 old females in phosphate-buffered saline (PBS). Salivary glands were sonicated using a Branson 450 sonicator with a 3/8 inches tip. Tube containing salivary glands in PBS were submerged in a beaker with water and placed adjacent to the submerged sonicator flat tip. The sonicator was set to 40% duty cycle, output 4 and the tube with glands received 20 pulses and then placed on ice for 1 min. The procedure was repeated 4 times and the tube with salivary glands was then centrifuged at 21,000 g for 3 min and the supernatant (SGH) was kept on ice and used throughout the study. SGH and recombinant proteins were routinely measured by bicinchoninic acid (BCA) protein assay kit (Pierce) following the manufacturer's protocol.

**Neutrophil isolation**. C57BL/6 bone marrow cells were harvested by flushing both femurs with Hank's balanced salt solution (HBSS) and neutrophils were isolated with a negative magnetic kit (MACS, Miltenyi Biotec) as recommended by the manufacturer. Neutrophils were resuspended in HBSS supplemented with 0.1% bovine serum albumin (BSA, Sigma).

Human and canine neutrophils were isolated from peripheral blood of healthy donors by a density gradient (Ficoll-Histopaque, GE). The layer containing neutrophils was subjected to red blood cells hypotonic lysis. Neutrophils were resuspended in HBSS supplemented with 0.1% BSA.

**Transwell assay**. HTS Transwell—96-well plates with 3 µm pore size polycarbonate membrane was purchased from Corning (#3386) and used to test the in vitro saliva chemoattractant activity. Neutrophils ($2.5 \times 10^5$ cells) were added to the top chamber of the transwell plate and allowed to set for 15 min prior to addition of chemoattractants. HBSS/0.1% BSA (control), *P. duboscqi* and *L. longipalpis* SGH or recombinant sand fly salivary proteins were added to the bottom chamber. After 3 h at 37° C/5% CO2 atmosphere, the bottom chamber content was recovered, centrifuged and neutrophil pellet was kept at −70 °C. Cell pellet was then thawed and a working solution of CyQuant (1:400 dilution in cell lysis buffer provided by the kit) dye was added to determine the cell number in the bottom compartment as directed by the manufacturer. The fluorescence of the samples was measured in a Spectramax Gemini XPS fluorescence microplate reader (Molecular Devices, Menlo Park, CA) with 485 excitation and 530 nm emission wavelengths. To test the directionality of neutrophil migration, the same concentration of SGH was added to both top and bottom chambers of the Transwell plate to disrupt the gradient.

To check the nature of saliva chemoattractant, *P. duboscqi* SGH (from ten pairs of salivary glands) was subjected to treatment with proteinase K (100 µg/mL) for 30 min at room temperature. Proteinase K activity was stopped by heating samples at 96 °C for 3 min. The control SGH which was not treated with proteinase K was also incubated at 96 °C for 3 min. In parallel, SGH (from ten pairs of salivary glands) were heated at 70 °C for 30 min or treated on ice with ethanol (final concentration of the ethanol was 90% v:v) for 1 h. Samples were centrifuged at 9300 *g* for 10 min and both supernatant and pellet were recovered. The ethanol from the collected supernatant was evaporated by centrifugation on a SpeedVac system and then resuspended in HBSS. Heated samples were brought to room temperature before use. For GPCR, PI3K and calcium influx blockage, neutrophils were pretreated for 1 h with pertussis toxin (Sigma, 2 µg/mL), or for 30 min with Wortmannin (1 µM; Sigma-Aldrich) and BAPTA-AM (10 µM, Calbiochem) at 37 °C/5% CO2 atmosphere.

**EZ-TAXIScan in vitro migration assay**. The EZ-TAXIScan chamber (Effector Cell Institute, Tokyo, Japan) was assembled as described by the manufacturer and filled with RPMI/0.1% BSA. All glass coverslips were ultrasonicated and washed before use and the coverslips and chips were coated with 2.5 µg/mL Fibronectin (Sigma) at room temperature for 1 h. Neutrophils (1 µL, $1 \times 10^6$/mL) were added to the lower reservoir of each of the six channels and allowed to line up by removing 18 µL of buffer from the upper reservoir. SGH and recombinant proteins were all diluted in RPMI/0.1% BSA. Two microliters of RPMI/0.1% BSA (control), fMLP (100 nM), *P. duboscqi* or *L. longipalpis* SGH and recombinant salivary proteins were then added to the upper reservoir and neutrophil migration (at 37 °C, in a humidified environmental chamber) in each of the channels was captured sequentially every 15 s for 3 h using a 10× lens. Neutrophil migration analysis was conducted with Imaris software. Single cells were automatically tracked over time and the tracks were then manually fixed. Cell tracks were subsequently used to compute the chemotaxis index and speed for each tracked cell. The chemotaxis index was calculated by dividing the track displacement by the track length. The track displacement is the distance between the first and last position and the track length is the total length of displacements within the track. The track speed was computed by dividing the track length by the time between first and last object in the track.

**Cloning and expression of *P. duboscqi* and *L. longipalpis* salivary proteins**. Cloning, expression and purification of recombinant *P. duboscqi* and *L. longipalpis* salivary glands were performed as described[54]. The PCR product was cloned into the VR2001- TOPO vector as previously described, sequenced and sent to the Protein Expression Laboratory at NCI-Frederick (Frederick, Maryland) for expression in HEK-293F cells commercially purchased from ThermoFisher scientific.

**Production of antibodies to recombinant salivary proteins**. Ten micrograms of recombinant PduM35 and PduM10 mixed (1:1 volume) with MagicTM Mouse Adjuvant (Creative Diagnostics, Shirley, NY) were injected intradermally in the ear of Balbc mice every 2 weeks for a total of 3 immunizations. Fourteen days after last immunization, serum was collected, and total IgGs were purified with the MelonTM Gel IgG Spin Purification Kit (ThermoFisher Scientific) following manufacturer's instructions.

**Western blot**. *P. duboscqi* SGH (five pairs of salivary glands) or recombinant proteins PduM35 and PduM10 (500 ng of each protein) were separated in a 4–12% Bis-Tris NUPAGE polyacrylamide gel. After electrophoresis, proteins were transferred to a nitrocellulose membrane (iBlotTM transfer stack, ThermoFisher Scientific). The membranes were incubated overnight with 5% nonfat dry milk (BioRad) diluted in TBST. After blocking, the membranes were incubated with immunized mice sera containing antibodies anti-PduM35 and PduM10 (1:1000 dilution) for 2 h at room temperature. After primary antibodies incubation, the membranes were washed 5 times with TBST and then incubated with alkaline phosphatase goat anti-mouse (Promega, Cat #S372B, Polyclonal, Lot # 312817; 1:10,000) for 1 h at room temperature. After 5 times washing with TBST, the

membranes were incubated with Western Blue—substrate for Alkaline Phosphatase (Promega) and exposed for 5 min until the bands were visible.

**Structure modeling of neutrophil-chemotactic protein**. PduM10 and PduM35 were modeled using the I-TASSER server with input of the amino acid sequences of the two proteins minus the predicted signal sequences. As expected, the top scoring modeling template was LJM11, a homolog of these proteins from *L. longipalpis*. The structural alignment algorithm TM produced a score of 0.978 with a root mean squared deviation (rmsd) of 0.67 Å for PduM10 and a TM score of 0.985 with a rmsd of 0.51 for PduM35. The modeled region covered essentially the entire molecule and the amino acid identities for both proteins with LJM11 was 0.496.

**In vivo neutrophil recruitment assay**. C57BL6 mice ears were exposed to the bite of a single *P. duboscqi* or injected with HBSS, SGH (from one pair of salivary glands) or with different concentrations of recombinant salivary proteins. All reagents were diluted in the same HBSS used for control and samples were injected in a volume of 10 µl. For some experiments, SGH was incubated with antibodies anti-PduM10 and PduM35 (1:100 dilution) for 30 min at room temperature before injection. After different time points, mice were euthanized and the two sheets of ear dermis were separated, deposited in PBS containing 0.2 µg/mL Liberase CI purified enzyme blend (Roche Diagnostics Corp.), and incubated for 1 h at 37 °C. Digested tissue was placed in a grinder and processed in a tissue homogenizer (Medimachine; Becton Dickenson). Tissue homogenates were filtered using a 30 µm Filcon filters (BD). The resulting single-cell suspensions were stained for Ly6C (BD Pharmingen™, Cat #553104, Clone AL-21, Lot #7067529; 1:200; FITC), Ly6G (BD Pharmingen™, Cat #551461, Clone 1A8, Lot #45938; 1:500; PE), CD11b (BD Pharmingen™, Cat #561098, Clone M1/70, Lot #31688; 1:500; PE-Cy7), and with the Fixable Yellow Dead Cell Stain Kit (Invitrogen), after being incubated with anti-Fc (TruStain FcX™ Rat anti-mouse CD16/32 Antibody, BioLegend, Cat #101319, Clone 93, Lot #B276722; 1:25) antibodies to block unspecific binding for 30 min. Data were analyzed on a MACSQuant flow cytometer (Miltenyi Biotec). Cells were acquired based on forward and side scatter and data analyzed with FlowJo Software 4.3.

**Intravital multiphoton imaging and analysis**. IVM was performed as previously described[55]. Briefly, images were acquired on a Leica DMi8 inverted 5-channel confocal microscope (Leica Microsystems) equipped with MaiTai and InSight DeepSea lasers (Spectra Physics). Ears were immobilized and immersed in lubricating jelly. Images were acquired with a 25× inverted objective (numerical aperture = 1). Images were obtained at 880 nm for eGFP-LysM (neutrophils) and collagen (dermis)—second harmonic generation. Emitted fluorescence was collected with ultra-sensitive hybrid detectors (HyD). Wavelength separation was accomplished with a 495 nm dichroic mirror followed by emission filters of 460/50 nm bandpass and 525/50 nm bandpass for SHG and GFP; a longpass filter of 560 nm; and 610/60 nm bandpass, a 650 nm longpass, a 685/50 bandpass for imaging far-red fluorophores. For movies 2 × 2 mosaic images were acquired with 0.75 zoom, a 5-µm z step for a depth of 100-µm every 2 min. Images and movies were analyzed using Imaris software (Bitplane). Maximum Intensity Projections (MIPs) were processed from Z stacks. The MFI of the GFP channel was calculated from MIPs using Fiji-nojre 2017 edition running under image j (version 1.52).

**Neutrophil depletion**. C57BL/6 mice were injected with 1 µg anti-Ly6G antibody (Rat InVivoMAb anti-mouse Ly6G, BioXCell, Cat #BE0075-1, Clone 1A8, Lot # 5002/1013) intraperitoneally 1 day before parasite infection. To check the efficiency and specificity of depletion, C57BL/6 mice ears were injected with KC (100 nM) and after 1 day we proceeded with Flow cytometry staining with anti-mouse Ly6G and Ly6C (BD Pharmingen, Cat #557661, Clone RB6-8C5, Lot # 6077709; 1:200; APC-Cy7), CD11b (clone M1/70; PE-Cy7; BD), and with the Fixable Yellow Dead Cell Stain Kit (Invitrogen), after being incubated with anti-Fc (CD16/32) antibodies to block unspecific binding for 30 min. Data were analyzed on a MACSQuant flow cytometer (Miltenyi Biotec). Cells were acquired based on forward and side scatter and data analyzed with FlowJo Software 4.3.

**Ear lesions measurement and parasite load**. Metacyclic promastigotes of *L. major* (500 parasites) in PBS or in association with recombinant salivary proteins (0.25 µM) or SGH (from one pair of salivary glands) were inoculated intradermally into both ears' dermis. Lesion progress was observed weekly by measuring ear thickness using a vernier caliper (Mitutoya America Corporation, Aurora, IL). Parasite load was determined using a limiting dilution assay. Mice were euthanized and the two sheets of ear dermis were separated, deposited in PBS containing 0.2 µg/mL Liberase CI purified enzyme blend (Roche Diagnostics Corp.), and incubated for 1 h at 37 °C. Digested tissue was placed in a grinder and processed in a tissue homogenizer (Medimachine; Becton Dickenson). Tissue homogenates were filtered using a 30 µm Filcon filters (BD). The resulting single-cell suspension was washed in PBS and resuspended in Schneider's medium supplemented with 10% heat-inactivated FBS, 100 U/mL penicillin, 100 µl/mL streptomycin and 2 mM l-glutamine and seeded in 96-well plates containing blood agar (Novy-Nicolle-McNeal). The number of viable parasites was determined from the highest dilution at which promastigotes could be found after 14 days of culture at 26 °C.

**Multiple sequence analysis**. For multiple sequence alignment analysis, PduM10, PduM35, LJM11, and LJM17 had their predicted signal peptide signal (SignalP-5.0 server[56]) removed, and resulting protein sequence entered into a Basic Local Alignment Search tool (BLAST)[57] against NR and TSA databases. We selected the five most similar homolog sequences (based on the e-value) for each sand fly species. The cut-off to exclude a homolog was an e-value above $1^{-10}$. Multiple sequence alignment and identity/similarity matrix were constructed on MacVector v15.5.3 with MUSCLE[58] using PAM 200 profile. Black shading represents identical amino acids, light gray shading represents similar amino acids.

**Statistical analysis**. Data were analyzed using GraphPad Prism version 8 for Mac. In all statistical tests, the reported P values are two-tailed, with $P < 0.05$ considered significant. Details of the used tests are included in the figure legends.

**Reporting summary**. Further information on research design is available in the Nature Research Reporting Summary linked to this article.

## Data availability
The data supporting the findings of this study are available within the paper or from the corresponding authors upon request. Please follow the links below to access the latest version of the BLAST tool (https://blast.ncbi.nlm.nih.gov/Blast.cgi) or the NR (https://ftp.ncbi.nlm.nih.gov/blast/db/) and TSA (https://www.ncbi.nlm.nih.gov/Traces/wgs/?view=TSA) databases. Source data are provided with this paper.

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

## Acknowledgements

We would like to thank the Twinbrook 3 animal facility for all the technical assistance provided. Special thanks to our Insectary team at the Laboratory of Vector and Malaria Research. A.B.G.C. and W.C. fellowships were partially supported by CNPq—Science without Borders Program (Brazil). I.W. and J.O. fellowships were supported by CAPES and CNPq (Brazil), respectively. This research was supported by the Intramural Research Program of the NIH, National Institute of Allergy and Infectious Diseases.

## Author contributions

A.B.G.C., J.B., H.D.H., S.K., J.G.V., and F.O. designed the research. A.B.G.C. performed the recombinant protein purification. A.B.G.C., J.P.S., W.C., X.W., I.W., J.O., and T.D.S. performed the neutrophil isolation, chemotaxis experiments, image acquisition and *Leishmania* infection experiments. C.M. dissected and provided sand fly salivary glands. A.B.G.C., J.P.S., and H.D.H. performed the in vivo migration imaging experiments. J.F.A. carried out the structural analysis. A.B.G.C., J.B., J.F.A., H.D.H., S.K. J.G.V. and F.O. wrote the manuscript.

## Funding

## Competing interests

The authors declare no competing interests.
