## [Peer Review File · Nature Communications]

REVIEWER COMMENTS

Reviewer #1 (Remarks to the Author):

A very nice piece of work.

This manuscript presents data to support the hypothesis that sandfly proteins are chemotactic for neutrophils. They identify specific proteins that are responsible for this effect and present in vivo data supporting a role for neutrophil accumulation at the bite site. Moreover, that this chemotactic response for neutrophils benefits the parasites, enhancing infection and parasite survival.

The concept of innate immunity promoting vector borne disease via neutrophil recruitment is not novel per se, as the authors cite in the discussion. Also, data from *Leishmania* has also previously identified neutrophil swarming response to sandfly saliva and the concept of benefit to the parasite via a “Trojan horse” type mechanism.

The present paper takes these concepts forward by identifying the proteins (at least some of them) that are responsible. Modelling studies suggest that the sandfly proteins do not bear close sequence or structural resemblance to a major mammalian chemokine for neutrophils.

Overall, this is a well-designed, carefully conducted and rigorously analysed set of experiments. The data is strong and supportive of their general hypotheses. I believe their conclusions are well justified, subject to certain caveats, and the data does now open up avenues for future exploitation of the specific proteins they have identified.

There are a number of points that the authors may like to consider.

In Figure 1, the data suggests that neutrophils from different host species respond differently to salivary gland extracts from different sandfly vectors. As Leishmaniasis is a zoonosis, this is potentially very interesting. Is it possible that vectors have evolved a “battery” of molecules to cover many hosts or could it help explain in part the differences between parasite species and their success of infection of particular hosts?

In Figure 2 the data presented from proteins from the two different vectors is used in assays with either murine or human neutrophils. Is there a particular reason for this?

In figure 3 a comparison is made between the structure of yellow proteins and CXCL1. Other biomolecules such as leukotrienes and C5a are potent neutrophil chemoattractants (see <https://www.pnas.org/content/109/46/E3177/1>). Have the authors compared their proteins to molecules such as these to look for structural or sequence similarities?

Also, going forward as the authors make a comparison specifically with CXCL1 it would be informative to compare neutrophil migration between equi-molar concentrations of the yellow proteins and CXCL1 to help gauge relative “potency”. I know the positive control was fMLP, but perhaps the chemokine would be a good/better comparison as this is the focus of the comparison.

Also in supplementary video one – there seems to be clear difference in the mode of movement/shape of cells that are migrating as compared to the other videos - the cells seem to move the cell body forward but not detach at the rear of the cell anyway as near as rapidly

as they seem to do on other successful migrations. Could the authors comment on this. Was this a one off observation and do the authors think it means anything?

On figure 3, C & D could you specify the time point looked at in the legend.

Reviewer #2 (Remarks to the Author):

In this paper, authors report a discovery of a neutrophil chemotactic factor in the sand fly saliva. The data show that two new proteins rPduM10 and rPduM35. rPduM10 elicits neutrophil chemotaxis in dose dependent manner. Furthermore, the findings show that both proteins also induce neutrophil migration in vivo upon injection into ear dermis. Overall, this is well-written paper that reports interesting findings but its descriptive as it stands. There are also several important issues that to be addressed to improve the scope and impact of the work.

Comments:

1. The work is descriptive and lacks mechanistic studies. Although authors identify new chemotactic proteins in sand fly saliva, they do not determine how these proteins are inducing chemotaxis. What is the direct effect of proteins on neutrophils? Do they activate GPCR signaling, AKT activation and actin polymerization? What is the receptor on neutrophils for these proteins? Do they induce IL-8 secretion from keratinocytes in vivo? Such mechanistic insights will improve the scope of the work.

2. What are the physiological levels of these proteins in sand fly saliva? How much will injected during a sand fly bite in natural infection? Are the concentrations used for in vitro and in vivo experiments physiological or pharmacological? Several other proteins in sand fly saliva such as rLJM17 and rLJM11 are also chemotactic. Maxadilan found in the saliva is also a vasodilator and could contribute to chemotaxis in vivo. Furthermore, sand fly saliva also contain bacteria, which can contribute to chemotaxis by producing fMLP and/or by inducing production of IL-8/KC from keratinocytes. Given that multiple other mechanisms could be involved neutrophil chemotaxis, are rPduM10 and rPduM35 redundant in vivo during natural transmission?

3. Authors show that rLJM17 and rLJM11 also induce chemotaxis. What are their contributions to chemotaxis in vivo? Do they induce chemotaxis through mechanisms that are similar to rPduM10 and rPduM35?

Point-by-point response to the reviewers comments

We thank both reviewers for the careful examination of the manuscript. Please see below our answers in blue to each of the points raised by the reviewers. We have new data showing that the recombinant yellow proteins are acting through a G-protein-coupled receptor-dependent manner. We also have compared the structure of the yellow proteins to CXCL8 and C5a. We have added two new supplementary figures, one that shows the protein alignment of all the available sand fly salivary yellow proteins and their identical and similar amino acids, and the other showing that *L. longipalpis* LJM11/17 combination attracts neutrophils in vivo similar to *P. duboscqi* PduM10/35.

REVIEWER COMMENTS

Reviewer #1 (Remarks to the Author):

A very nice piece of work.

This manuscript presents data to support the hypothesis that sandfly proteins are chemotactic for neutrophils. They identify specific proteins that are responsible for this effect and present in vivo data supporting a role for neutrophil accumulation at the bite site. Moreover, that this chemotactic response for neutrophils benefits the parasites, enhancing infection and parasite survival.

The concept of innate immunity promoting vector borne disease via neutrophil recruitment is not novel per se, as the authors cite in the discussion. Also, data from *Leishmania* has also previously identified neutrophil swarming response to sandfly saliva and the concept of benefit to the parasite via a “Trojan horse” type mechanism.

The present paper takes these concepts forward by identifying the proteins (at least some of them) that are responsible. Modelling studies suggest that the sandfly proteins do not bear close sequence or structural resemblance to a major mammalian chemokine for neutrophils.

Overall, this is a well-designed, carefully conducted and rigorously analysed set of experiments. The data is strong and supportive of their general hypotheses. I believe their conclusions are well justified, subject to certain caveats, and the data does now open up avenues for future exploitation of the specific proteins they have identified.

We thank the reviewer for all the positive comments about our manuscript. We also agree that we have taken this concept forward by identifying and characterizing the protein responsible for this effect.

There are a number of points that the authors may like to consider.

In Figure 1, the data suggests that neutrophils from different host species respond differently to salivary gland extracts from different sandfly vectors. As Leishmaniasis is a zoonosis, this is potentially very interesting. Is it possible that vectors have evolved a “battery” of molecules to cover many hosts or could it help explain in part the differences between parasite species and their success of infection of particular hosts?

This is an interesting question and we acknowledge the possibility that vectors evolved their salivary molecules to adapt to different hosts. In fact, *P. duboscqi* sand flies are mostly found in close proximity to rodent burrows and consequently will feed more frequently on rodents, while *L. longipalpis* seems to have an eclectic feeding preference including chickens, cows, dogs and humans as indicated by several studies on sand fly bloodmeal analysis and feeding preference. This fits well with our in vitro transwell results for whole saliva neutrophil chemotactic attraction.

In Figure 2 the data presented from proteins from the two different vectors is used in assays with either murine or human neutrophils. Is there a particular reason for this?

Thank you for this observation. The reason relates to your comment above where after we learned that *P. duboscqi* saliva was more effective in recruiting murine neutrophils and *L. longipalpis* saliva recruited more efficiently human neutrophils in the transwell assays (Figure 1), we decided to perform the transwell assays with the following pairs: i) *P. duboscqi* SGH or proteins (rPduM10/35) with murine neutrophils and ii) *L. longipalpis* SGH or proteins (rLMJ11/17) with human neutrophils throughout the study.

We have added a sentence to further highlight it in the results section of the manuscript, lines 174 – 177 as follows:

We tested transwell migration for *P. duboscqi* recombinant proteins in murine neutrophils, while *L. longipalpis* recombinant proteins were tested in human neutrophils to maximize their migration potential according to data in Figure 1B-D.

In figure 3 a comparison is made between the structure of yellow proteins and CXCL1. Other biomolecules such as leukotrienes and C5a are potent neutrophil chemoattractants (see <https://www.pnas.org/content/109/46/E3177/1>). Have the authors compared their proteins to molecules such as these to look for structural or sequence similarities.

We thank the reviewer for the suggestions and we have added to figure 3G the structure of C5a and CXCL8 in addition to CXCL1. Leukotrienes are indeed strong inducers of neutrophil migration, however they are lipids and we showed in Fig. 2B that the sand fly chemoattractant had no lipid component. We also searched for structural similarities to other proteins in all the available databases and we did not find a match. We have modified the results lines 302-311 :

In comparison, the neutrophil recruiting chemokines CXCL1 or CXCL8 have a symmetrical dimeric structure containing a single six-stranded β -sheet made up of three strands from each monomer laid up against two α -helices also contributed by the two monomers (Fig. 3G). The structure of C5a, a chemotactic complement component, consists of a four α -helix bundle and also does not resemble the β -propeller structure of the yellow proteins (Fig. 3G). These data indicate that the neutrophil chemoattractant proteins, PduM10 and PduM35 bear no similarity to the structure of the vertebrate neutrophil-attracting proteins CXCL1, CXCL8 or C5a.

The reviewer's comments gave us the idea to include a protein alignment to highlight that the yellow family is very conserved throughout Old and New world sand fly species that evolved in different continents for millions of years. The identity between *P. duboscqi* and *L. longipalpis* yellows vary from 43.6 % to 48.8 %. These data can help us locate areas in this yellow family that may harbor the chemoattractant motifs. The level of conservation also hints that this chemoattractant function may be conserved across sand fly species. We have added one figure (Supplementary Figure 4A-B) that shows the protein alignment of the tested recombinant *P. duboscqi* and *L. longipalpis* yellows (A) and an alignment of all available salivary yellow proteins from sand flies (B). Of note, no matches with mammal proteins were found in all searched databases.

We have added the paragraph below to text lines 187 - 193 :

Amino acid identity between *P. duboscqi* rPduM10 and rPduM35, and *L. longipalpis* rLJM17 and rLJM11 yellow proteins range from 43.6 % to 48.8 %. Alignment shows several conserved areas across these proteins (Supplementary Fig. 4A) that were also maintained across all described salivary yellow proteins from sand flies (Supplementary Fig. 4B), likely indicating a conserved function for this protein family.

Also, going forward as the authors make a comparison specifically with CXCL1 it would be informative to compare neutrophil migration between equi-molar concentrations of the yellow proteins and CXCL1 to help gauge relative "potency". I know the positive control was fMLP, but perhaps the chemokine would be a good/better comparison as this is the focus of the comparison.

We thank the reviewer for the comment. We want to point out that our comparison with CXCL1 (and now CXCL8 and C5a) was to demonstrate that the yellow chemotactic function is divorced from a bona fide chemokine signature.

To address the reviewer's points, we compared the migration of human neutrophils towards CXCL8 (or IL-8) or LJM11 and LJM17 yellow proteins in a dose-dependent response transwell assay (Fig. A) and we observed that the commercially available recombinant IL-8 recruits around 5 times and 10 times more neutrophils than LJM11 and LJM17 yellow proteins, respectively, at the same molarity (Fig. A). WKYMVm is a peptide agonist of formyl peptide receptors (also acting through G-protein-coupled receptor) and migration of murine neutrophils was compared to rPduM10 and rPduM35 proteins (Fig. B). This peptide was twice as efficient as *P. duboscqi* rPduM10 and rPduM35 proteins in recruiting murine neutrophils.

Also in supplementary video one – there seems to be clear difference in the mode of movement/shape of cells that are migrating as compared to the other videos - the cells seem to move the cell body forward but not detach at the rear of the cell anyway as near as rapidly as they seem to do on other successful

migrations. Could the authors comment on this. Was this a one off observation and do the authors think it means anything?

We agree with the observations of the reviewer and we think this is probably related to the use of whole salivary gland homogenate and gradient formation in the EZ-TAXIScan slides. Since sand fly saliva contain around 35 distinct proteins, neutrophils can take longer to sense the yellow salivary chemoattractant gradient. We see that this effect is absent when the purified recombinant yellow proteins were tested compared to positive control.

We have added the information below to the results section, lines 111-114 as reads below :

Of note, despite migrating at a similar speed, we observed that neutrophils took longer to sense the SGH gradient, possibly due to the complex mixture of proteins (around 35 distinct proteins) present in sand fly saliva.

On figure 3, C &D could you specify the time point looked at in the legend.

We apologize for this oversight, the time point is 12 hours and it has been added to the respective figure legend.

Reviewer #2 (Remarks to the Author):

In this paper, authors report a discovery of a neutrophil chemotactic factor in the sand fly saliva. The data show that two new proteins rPduM10 and rPduM35. rPduM10 elicits neutrophil chemotaxis in dose dependent manner. Furthermore, the findings show that both proteins also induce neutrophil migration in vivo upon injection into ear dermis. Overall, this is well-written paper that reports interesting findings but its descriptive as it stands. There are also several important issues that to be addressed to improve the scope and impact of the work.

We thank the reviewer for the comments about our work.

Comments:

1. The work is descriptive and lacks mechanistic studies. Although authors identify new chemotactic proteins in sand fly saliva, they do not determine how these proteins are inducing chemotaxis. What is the direct effect of proteins on neutrophils? Do they activate GPCR signaling, AKT activation and actin polymerization? What is the receptor on neutrophils for these proteins? Do they induce IL-8 secretion from keratinocytes in vivo? Such mechanistic insights will improve the scope of the work.

We thank the reviewer for their insights and questions. The process of identifying a novel molecule is not merely descriptive, and it requires significant effort and expertise in different areas of biochemistry, protein chemistry and molecular biology. However, we acknowledge that adding mechanistic information will improve the presentation of our work. Therefore, we performed experiments to test if these novel proteins are exerting their function using G-protein-coupled receptors, PI3K activation and calcium influx. Our new data indicates that neutrophil migration induction by the sand fly PduM10/35 or LJM11/17 proteins is dependent on G-protein-coupled receptor activation and calcium influx. We observed that PI3K is also relevant for PduM10/35 directed migration, but not statistically significant to LJM11/17 migration. This is now presented in Fig. 2J for PduM10/35 and Fig. 2K for LJM11/17. We have added the following text below to the results and the discussion.

At the Results section lines 203 - 211 :

We then tested if the migration towards yellow proteins was dependent on with neutrophil G-protein-coupled receptors (GPCR), phosphatidylinositol 3-kinases (PI3K) activity, or calcium influx. When neutrophils were pre-treated with either pertussis toxin (PTx) (blocking GPCR-mediated migration) or BAPTA-AM (which chelates calcium), their migration toward PduM10/35 (**Fig. 2J**) and LJM11/17 (**Fig. 2K**) was abrogated in the transwell assay. Interestingly, pre-treatment of neutrophils with wortmannin, a pan-

PI3K inhibitor, abolished migration to PduM10/35 (**Fig. 2J**), but some migration towards LJM11/17 persisted (**Fig. 2K**).

At the Discussion section lines 412 - 418:

These sand fly salivary yellow protein chemoattractants follow a classical pathway of GPCR-activated neutrophil migration²², that is dependent on calcium influx and, for PduM10/35, on PI3K signaling^{23,24}. Unexpectedly, the inhibition of PI3K by wortmannin was not sufficient to abrogate migration towards LJM11/17. Further studies with PI3K-deficient neutrophils and other PI3K selective inhibitors will corroborate if LJM11/17 chemotaxis is PI3K independent, similar to what has been observed in LPS-primed humans cells²⁵.

2. What are the physiological levels of these proteins in sand fly saliva? How much will be injected during a sand fly bite in natural infection? Are the concentrations used for in vitro and in vivo experiments physiological or pharmacological?

We thank the reviewer for this comment, and it is important to stress that we believe we are working under physiological conditions.

In a previous work, we estimated that sand flies can deliver 70% of their salivary proteins content while they are feeding (Kato et al. *Journal of Experimental Biology* 2007 210: 733-740; doi: 10.1242/jeb.001289). We also measured the total protein in salivary gland homogenate of sand flies and it is around 1000 nanograms. We have estimated by transcriptomics and proteomics analysis that the yellow salivary proteins represent 30% of the total proteins present in sand fly salivary glands. Based on the above, we calculated that the amount of Yellow proteins that can be injected during a sand fly bite is around 200 ng.

We designed the experiments to inoculate amounts of salivary proteins that would be under physiological conditions based on these calculations. For example: 100 nanograms of the yellow protein in 10 microliters (injection volume) equals to 0.25 micromolar. Another point to consider is that the insect will deliver its saliva in a focal microenvironment, therefore, salivary proteins will be much more concentrated within the bite site.

We have added a paragraph to the results section: lines 265-269

Of note, one sand fly has approximately 300 ng of yellow proteins in its salivary glands. Here, the tested recombinant protein concentration of 0.25 μM in the injection volume of 10 μL (= 100 ng) is under the physiological range of yellow proteins injected during a sand fly bite.

Several other proteins in sand fly saliva such as rLJM17 and rLJM11 are also chemotactic. Maxadilan found in the saliva is also a vasodilator and could contribute to chemotaxis in vivo. Furthermore, sand fly saliva also contain bacteria, which can contribute to chemotaxis by producing fMLP and/or by inducing production of IL-8/KC from keratinocytes. Given that multiple other mechanisms could be involved neutrophil chemotaxis, are rPduM10 and rPduM35 redundant in vivo during natural transmission?

We agree with the reviewer and we also think that neutrophil chemotaxis may be a redundant system induced by a sand fly bite during transmission, where saliva (yellow proteins), bacteria egested from the sand fly gut and the *Leishmania* parasite will induce neutrophil chemotaxis that will ultimately benefit the parasite to get established. Nevertheless, regarding to the effect of sand fly saliva, we show in Fig. 3D and Fig. 4D-E that when the yellow proteins in *P. duboscqi* saliva is neutralized the chemotactic effect of saliva and its consequences on *Leishmania* parasite loads are abolished. Maxadilan is a very potent vasodilator but we have no evidence that this protein has chemotactic activity.

3. Authors show that rLJM17 and rLJM11 also induce chemotaxis. What are their contributions to chemotaxis in vivo? Do they induce chemotaxis through mechanisms that are similar to rPduM10 and rPduM35?

We thank the reviewer for this comment. We now have tested a combination of LJM11/17 in vivo. We observe that the yellow proteins from *L. longipalpis* also

attract neutrophils to the site of injection (Sup. Fig. 5). We have added this information to the results section lines 261-265 as follows:

Additionally, we tested a combination of 0.25 μ M of *L. longipalpis* yellow proteins LJM11 + LJM17 in vivo and observed a statistically significant neutrophil recruitment 12 hrs post-injection (Supplementary Fig. 5), corroborating in vitro data on the chemoattractant role of the yellow proteins from both *L. longipalpis* and *P. duboscqi* in vivo.

The reviewer's comments gave us the idea to include a protein alignment to highlight that the yellow family is very conserved throughout Old and New world sand fly species that evolved in different continents for millions of years. The identity between *P. duboscqi* and *L. longipalpis* yellows vary from 43.6 % to 48.8 %. These data can help us locate areas in this yellow family that may harbor the chemoattractant motifs. The level of conservation also hints that this chemoattractant function may be conserved across sand fly species. We have added one figure (Supplementary Figure 4A-B) that shows the protein alignment of the tested recombinant *P. duboscqi* and *L. longipalpis* yellows (A) and an alignment of all available salivary yellow proteins from sand flies (B). Of note, no matches with mammal proteins were found in all searched databases.

We have added the paragraph below to text lines 187 - 193 :

Amino acid identity between *P. duboscqi* rPduM10 and rPduM35, and *L. longipalpis* rLJM17 and rLJM11 yellow proteins range from 43.6 % to 48.8 %. Alignment shows several conserved areas across these proteins (Supplementary Fig. 4A) that were also maintained across all described salivary yellow proteins from sand flies (Supplementary Fig. 4B), likely indicating a conserved function for this protein family.

REVIEWER COMMENTS

Reviewer #1 (Remarks to the Author):

The authors have adequately addressed the issues that I raised.

Reviewer #3 (Remarks to the Author):

This is a very complex study comprising several experiments in-vivo as well in-vitro. The data are technically sound and results of all these experiments nicely support the hypothesis about the role of yellow proteins in neutrophil recruitment and the paper provides strong evidence for its conclusions. Results are novel, except for protein models, see below. The manuscript is important, especially to the scientists in the specific field of vector biology, and I like to congratulate authors to these interesting results.

Authors present models of *Phlebotomus duboscqi* yellow proteins as a novel result. However, models of *P. duboscqi* proteins were published already in PLoS One by Sima et al. 2016 „The diversity of Yellow-related proteins in sand flies“(models are in the Supplement). Please, refer to Sima et al. and explain if your models were made using different methods or software. Did you use the same template? Could you indicate any differences in protein structure between your models and those previously published?

I think that this study by Sima et al. 2016, describing the structure of more than 30 different yellow proteins of various sand fly species, should be cited also on line 296 and elsewhere in the text.

Based on the structure of yellow proteins, could you hypothesize which part of the sequence is important for chemotaxis? Similarly, could you explain/hypothesize about the mechanism of binding to neutrophils?

In Discussion I am missing more info about amine-binding properties of yellow proteins. For example, there is a recent study by Sumova et al. 2019 in Insect Biochem Mol Biol, showing that also *Phlebotomus* yellow proteins possesses the amine binding activities (Xu et al. demonstrated previously for a single *Lutzomyia* species).

Yellow proteins are also well-known markers of exposure, as demonstrated in various sand flies and summarized in review by Lestnova et al (2017). Moreover, Willen et al. recently developed a rapid diagnostic test based on recombinant yellow protein of *Phlebotomus perniciosus*. I think that it would be good to mention this interesting info about yellow proteins in Discussion.

In Results, lines 265-267, there is an interesting and important note about the quantity of yellow proteins in salivary glands of sand flies. However, do you mean yellow proteins of *P. duboscqi* or *L. longipalpis*, or both species? There is a difference between protein content of salivary glands of these two species, right? In addition, please explain if this is your result (then I would expect more details how did you measure it) or refer to the source of this information.

In coinjection experiments (page 18) I appreciate that authors inoculate 500 metacyclic parasites. It would be good to explain to non-experts that this number is within the natural

range of parasite numbers transmitted by sand flies (optimally plus the reference).

I appreciate also the effort spent during testing the effect of yellow proteins on exacerbation of *L. major* Infection. Limited dilution assay is quite laborious technique. Please explain why you did not use Q-PCR, the method which is currently widely used for parasite quantification.

Line 450: please change the reference, Rogers et al. is No. 33 (No 32 is Li et al. on different topic).

Line 453 and the list of References: Ref. 34 should be replaced by different Giraud et al. The correct reference is the paper published in PLoS Pathogens (2018) and entitled "Leishmania proteophosphoglycans regurgitated from infected sand flies....."

In Materials and Methods (line 510) please give more info about sonication (e.g. about the type of sonicator).

Fig. 1. In legend please explain „fMLP“

Fig. 2. In legend please explain „fMPL“ , „HBSS“ and „Ptk“

Suppl. Fig. 2: LJM111 protein is included (but nothing about it is in the text). Why? Did you study the chemotactic activity of LJM111 and did you find any?

Suppl. Fig. 3: Again, there is also the experiment with LJM111. If I understand well, a relatively small number of neutrophils was used in the assay with this proteins (in contrast to other proteins tested). Could you explain this, please?

Suppl. Fig. 4: Please mention what is the difference against alignments published by Sima et al. Which "new" sand fly species did you include?

I was asked to comment further on previous concerns of an absent reviewer 2 and the authors' response. Below is my opinion. In comments 1 and 2, I copy first the original questions of the Reviewer 2 and my text follows in the second paragraph.

Referee 2, comment No. 1

The work is descriptive and lacks mechanistic studies. Although authors identify new chemotactic proteins in sand fly saliva, they do not determine how these proteins are inducing chemotaxis. What is the direct effect of proteins on neutrophils? Do they activate GPCR signaling, AKT activation and actin polymerization? What is the receptor on neutrophils for these proteins? Do they induce IL-8 secretion from keratinocytes in vivo? Such mechanistic insights will improve the scope of the work.

Authors added experiments to test if these novel proteins are exerting their function using G protein-coupled receptors, PI3K activation and calcium influx. This is, of course, a significant improvement but I am not sure if it would be satisfactory for the reviewer. In my opinion, authors did not respond to most of his questions above. I understand that these are difficult questions but they should not be left without an answer. In my opinion, most of these questions (e.g. about the receptor on neutrophils or how these proteins are inducing

chemotaxis) should be answered at least in the letter.

Comment No. 2.

What are the physiological levels of these proteins in sand fly saliva? How much will be injected during a sand fly bite in natural infection? Are the concentrations used for in vitro and in vivo experiments physiological or pharmacological?

I appreciate detailed explanation given by the authors in their response. However, this explanation must appear also into the Results or Discussion. The paragraph added by the authors to the results section, lines 265-269, is not enough; it is very short and general. It lacks any reference and specification of sand fly species.

Comment No. 3.

In my opinion, this comment is answered satisfactorily. However, the added figure (Supplementary Figure 4) showing the protein alignment seems to be quite similar to alignment previously published by Sima et al. (2016). Would be good to comment this and explain differences.

Answer to reviewers

Reviewer #1 (Remarks to the Author):

Q. The authors have adequately addressed the issues that I raised.

Answer: We thank the reviewer for acknowledging and accepting the changes we made to the manuscript.

Reviewer #3 (Remarks to the Author):

Q. This is a very complex study comprising several experiments in-vivo as well in-vitro. The data are technically sound and results of all these experiments nicely support the hypothesis about the role of yellow proteins in neutrophil recruitment and the paper provides strong evidence for its conclusions. Results are novel, except for protein models, see below. The manuscript is important, especially to the scientists in the specific field of vector biology, and I like to congratulate authors to these interesting results.

Answer: We thank the reviewer for pointing out the significance of the study, its novelty and that our data support our conclusions.

Q. Authors present models of *Phlebotomus duboscqi* yellow proteins as a novel result. However, models of *P. duboscqi* proteins were published already in PLoS One by Sima et al. 2016 „The diversity of Yellow-related proteins in sand flies“(models are in the Supplement). Please, refer to Sima et al. and explain if your models were made using different methods or software. Did you use the same template? Could you indicate any differences in protein structure between your models and those previously published?

Answer: We thank the reviewer for pointing this out. Our intention was not to indicate that the structure we are showing of the sand fly yellow is novel, but to make clear to the reader that the structure of yellow proteins from sand fly saliva are different to known mammalian chemokines, this was also pointed out by a reviewer in the previous revision. In fact, we added the structure of another mammalian chemokine to further stress this important point. Additionally, we wrote in the text and in the methods that this was modelled from the yellow protein from the sand fly *Lutzomyia longipalpis*. This modeling was performed by the co-author, John Andersen, who crystalized and solved the structure of the yellow protein from *L. longipalpis*. Based on the reviewer's comment I think it is appropriate to include the information regarding the reported models of yellow proteins in *P. duboscqi* and the reference. This is now added to the text of the manuscript and it now reads:

Lines 320-325, Main Text: "A similar structure was independently reported for these same yellow proteins(Sima et al., 2016). Models of Pdup1 (equivalent to PduM10) and Pdup2 (equivalent to PduM35) were very similar with PduM10 and Pdup1 showing a root-mean-square deviation (rmsd) of 0.20 Å over 304 Cα positions and PduM35 and Pdup2 showing a rmsd value of 0.21 Å over 307 Cα positions reinforcing accuracy of the models."

Q. I think that this study by Sima et al. 2016, describing the structure of more than 30 different yellow proteins of various sand fly species, should be cited also on line 296 and elsewhere in the text.

Answer: We agree with the reviewer and we have corrected this oversight and added the reference for this work in the text and in the references. Of note, in the new amino acid alignment of yellow proteins in supplementary figure 4B, we now include three new yellow proteins from sand fly species: *S. schwetzi*, *N. neivai* and *P. kandelakii*.

Q. Based on the structure of yellow proteins, could you hypothesize which part of the sequence is important for chemotaxis?

Answer: This is a very interesting question, but it is difficult to answer at this point for two reasons: 1) blood feeding arthropods in general use different salivary proteins and different motifs for their biological activities, meaning that when they were discovered most of these activities as well as their sequences were novel, 2) This is a novel chemoattractant and there is no precedent on the structure-function relationship to other known chemoattractant. Based on these facts, we cannot speculate on the nature of the receptor-ligand binding motifs. Current models for chemokine receptors take into account the small size and motifs of well know chemokines. Due to the novelty and uniqueness of these interactions, we think the best approach would be to obtain crystals of the protein–ligand complex and solve the structure. This is planned for a future study as it will take a considerable effort and time because we need to first identify the GPCR receptor in neutrophils that is critical for the chemotactic function induced by these two sand fly yellow proteins. Overall, what makes this study interesting is finding a novel chemoattractant activity from a salivary molecule that is known to have another biological activity (bioamine binding protein) and that can alter the response of a mammalian immune cell by inducing chemotaxis.

Q. Similarly, could you explain/hypothesize about the mechanism of binding to neutrophils?

Answer: This is also very difficult for the same reasons stated above. This will be a line of investigation that will require a few years of research by our as well as other groups.

Q. In Discussion I am missing more info about amine-binding properties of yellow proteins. For example, there is a recent study by Sumova et al. 2019 in Insect Biochem Mol Biol, showing that also Phlebotomus yellow proteins possesses the amine binding activities (Xu et al. demonstrated previously for a single Lutzomyia species).

Answer: I think this is a very good point by the reviewer and it is something we can expand more on this paper. This will definitely emphasize the relevance of this finding because the yellow proteins are known to bind biogenic amines and now, we are

characterizing a novel function for this family of proteins. We thank the reviewer for this suggestion and the text now reads:

Lines 460-465, Discussion: “More recently, it was shown that the yellow proteins from the genus *Phlebotomus* also have the function of biogenic amine binding proteins (Sumova et al., 2019) demonstrating this biological function is present in these relevant vectors of *Leishmania*. We have now shown a dual function of this abundant and ubiquitous sand fly family of yellow salivary proteins as a neutrophil chemoattractant that recruits cells to the bite site.”

Q. Yellow proteins are also well-known markers of exposure, as demonstrated in various sand flies and summarized in review by Lestinova et al (2017). Moreover, Willen et al. recently developed a rapid diagnostic test based on recombinant yellow protein of *Phlebotomus perniciosus*. I think that it would be good to mention this interesting info about yellow proteins in Discussion.

Answer: We agree with the reviewer and this will actually emphasize the broader impact of this family of proteins. We have added the following information in the discussion section:

Lines 465-470, Discussion: “Thus far, the presence of the yellow family of proteins has only been confirmed in saliva of sand flies. In fact, this family of salivary proteins have been characterized as biomarkers for sand fly exposure on humans and dogs (Abdeladhim, Kamhawi, & Valenzuela, 2014; Lestinova, Rohousova, Sima, de Oliveira, & Volf, 2017; Teixeira et al., 2010) and recently a diagnostic test was developed based on this family of sand fly salivary proteins using a recombinant protein from the sand fly *Phlebotomus perniciosus* (Willen et al., 2019).”

Q. In Results, lines 265-267, there is an interesting and important note about the quantity of yellow proteins in salivary glands of sand flies. However, do you mean yellow proteins of *P. duboscqi* or *L. longipalpis*, or both species? There is a difference between

protein content of salivary glands of these two species, right? In addition, please explain if this is your result (then I would expect more details how did you measure it) or refer to the source of this information.

Answer: The reviewer is correct as it was previously shown by Cerna et al (Cerna et al. 2002) that different sand flies will have different amounts of proteins in their salivary glands. Our calculations for the amounts of yellow proteins in salivary glands of sand flies are based on previous measurements of the amount of protein in salivary glands in our laboratory that are in agreement with previous and recent publications (Rogers et al, 2004, Prates et al, 2008; Ribeiro et al., 1989; Kato et al. 2007, Mondragon-Shem, 2020, Sumova et al, 2019) and the estimate of yellow salivary proteins delivered by sand flies recently reported by Sumova et al (Sumova et a 2019). Our objective in the text is mainly to inform the reader that we are using physiological amounts of the recombinant yellow proteins for the relevant experiments. We have added the following information to the text :

Lines 273-286, Main Text: Of note, the amount of proteins in a pair of salivary glands varies by sand fly species(Cerna, Mikes, & Volf, 2002; Lestinova et al., 2017), and a sand fly delivers 70-90 % of its salivary protein content while feeding(Kato, Jochim, Lawyer, & Valenzuela, 2007; Prates et al., 2008; Ribeiro, 1989). For *P. duboscqi* and *L. longipalpis*, a pair of salivary glands contains between 0.78 µg to 1 ug of protein (Cerna et al., 2002; Kato et al., 2007) and 0.5 ug to 1 ug of protein (Ferreira et al., 2016; Mondragon-Shem et al., 2020), respectively. Further, transcriptomic analysis indicates that the relative abundance of yellow proteins in the salivary glands of *L. longipalpis* and *P. duboscqi* is 16.0% (Valenzuela, Garfield, Rowton, & Pham, 2004) and 7.5% (Kato et al., 2006), respectively. Based on the above, and from a recently reported estimate of the amount and proportion of yellow salivary proteins present in sand flies (Sumova et al., 2019), we estimate there is approximately 70 ng of yellow proteins in *P. duboscqi* and approximately 120 ng of yellow proteins in *L. longipalpis* salivary glands. Therefore, the tested recombinant protein concentration of 0.25 µM in an injection volume of 10 µL

(= 100 ng) is within the physiological range of yellow proteins injected during a sand fly bite (Sumova et al., 2019).

Q. In coinjection experiments (page 18) I appreciate that authors inoculate 500 metacyclic parasites. It would be good to explain to non-experts that this number is within the natural range of parasite numbers transmitted by sand flies (optimally plus the reference).

Answer: We added this information in the text and added a reference. It now reads:

Lines 367-371, Main Text: “To determine the effect of the two neutrophil chemoattractant proteins on *Leishmania* infection, we injected 500 *Leishmania major* metacyclics, similar to the median number of parasites reported to be delivered by a sand fly bite (Kimblin et al., 2008), either alone or in conjunction with 0.25 μ M of recombinant rPduM10 and rPduM35.”

Q. I appreciate also the effort spent during testing the effect of yellow proteins on exacerbation of *L. major* Infection. Limited dilution assay is quite laborious technique. Please explain why you did not use Q-PCR, the method which is currently widely used for parasite quantification.

Answer: We routinely use Limited dilution assay (LDA) for animal tissue parasite quantification, because of its consistency and because it denotes the number of live replicating parasites. Q-PCR is available in the laboratory, but all things considered, we actually found Q-PCR to be more expensive and laborious compared to LDA in our setting.

Q. Line 450: please change the reference, Rogers et al. is No. 33 (No 32 is Li et al. on different topic).

Answer: We thank the reviewer for this observation. We have deleted reference 32, Li et al and replaced the reference with the appropriate reference Rogers et al.

Q. Line 453 and the list of References: Ref. 34 should be replaced by different Giraud et al. The correct reference is the paper published in PLoS Pathogens (2018) and entitled “Leishmania proteophosphoglycans regurgitated from infected sand flies.....”

Answer: We thank the reviewer for this observation. We have replaced the reference in endnote to: Giraud Et al, 2018 : PLoS Pathog 2018 Jan 19;14(1):e1006794. doi: 10.1371/journal.ppat.1006794

Q. In Materials and Methods (line 510) please give more info about sonication (e.g. about the type of sonicator).

Answer: We have added this information in material and methods:

Lines 547-556, Methods: “Salivary glands were dissected from 5-7 old females in phosphate-buffered saline (PBS). Salivary glands were sonicated using a Branson 450 sonicator with a 3/8 inches tip. Tube containing salivary glands in PBS were submerged in a beaker with water and placed adjacent to the submerged sonicator flat tip. The sonicator was set to 40% duty cycle, output 4 and the tube with glands received 20 pulses and then placed on ice for 1 minute. The procedure was repeated 4 times and the tube with salivary glands was then centrifuged at 21,000 g for 3 minutes and the supernatant (SGH) was kept on ice and used throughout the study. SGH and recombinant proteins were routinely measured by bicinchoninic acid (BCA) protein assay kit (Pierce) following the manufacturer’s protocol.”

Q. Fig. 1. In legend please explain „fMLP“

Answer: We thank the reviewer for pointing this out. We have added this information to Figure 1, Figure 2 and in the main text.

Lines 143-145, Main Text: “**(F-H)** Validation of sand fly salivary gland homogenate driven human neutrophil recruitment using EZ-TAXIScan assay. N-Formylmethionyl-leucyl-phenylalanine (fMLP) was used as a positive control.”

Lines 110-111, Main Text: “to N-Formylmethionyl-leucyl-phenylalanine (fMLP), the positive control (Supplementary Video 2).”

Q. Fig. 2. In legend please explain „fMPL“, „HBSS“ and „Ptk“

Answer: We thank the reviewer for pointing this out. We have added the following information to figure 2:

Lines 246-248, Fig. legend 2: “(G-I) Validation of yellow proteins driven human neutrophil recruitment in EZ-TAXIScan assay. fMLP (N-Formylmethionyl-leucyl-phenylalanine) was used as a positive control. Hanks' Balanced Salt Solution (HBSS) was used as our baseline control.”

Line 233, Fig. legend 2: “(A) Ten pairs of *P. duboscqi* salivary glands were treated or not with Proteinase K (Ptk) to digest salivary proteins.”

Q. Suppl. Fig. 2: LJM111 protein is included (but nothing about it is in the text). Why? Did you study the chemotactic activity of LJM111 and did you find any?

Answer: We thank the reviewer for pointing this out. We also did not write in the text the results for LJM111 and PduM34. The LJM111 protein was produced, as shown in supplementary Figure 2 and tested for chemotactic activity. We did not observe neutrophil recruitment from LJM111, LJM143 and PduM34 as shown in supplementary figure 3.

We added this information in the main text:

Lines 170-176, Main Text: “To identify the protein responsible for neutrophil chemotaxis in sand fly saliva, we chose to test the most abundant proteins in saliva of *P. duboscqi* and *L. longipalpis* sand flies: PduM10, PduM35, LJM11, LJM111, and LJM17 from the yellow family of proteins, PduM34 from the silk related family of proteins, and LJM143 from the Lufaxin family of proteins. We expressed these sand fly salivary proteins in HEK cells, purified them as endotoxin-free recombinant proteins (Supplementary Fig. 2)...”

Lines 180-184, Main Text: “...Only members of the yellow family of proteins from the saliva of both *P. duboscqi*, PduM10, PduM35) and *L. longipalpis* LJM11, and LJM17) exhibited chemotactic activity in the EZ-TAXIScan assays. We did not observe a chemotactic activity for the recombinant salivary proteins LJM111, LJM143 and PduM34 (Supplementary Fig. 3).”

Furthermore, because LJM111 belongs to the yellow family of proteins and LJM111 recombinant protein did not show chemotactic activity, we modified the abstract and discussion to indicate that we identified only members of the yellow family of proteins with this activity in the two sand flies tested, it now reads:

Lines 34-36, Abstract: “Here, we show that members of sand fly yellow salivary proteins induce *in vitro* chemotaxis of mouse, canine and human neutrophils in transwell migration or EZ-TAXIScan assays.”

Lines 441-448, Discussion: “Interestingly, LJM111, a third member of the yellow family of salivary proteins in *L. longipalpis*, did not show neutrophil chemotactic activity in our assays. LJM111 is phylogenetically distant from the yellow salivary protein LJM17 and less distant to LJM11 (Valenzuela et al., 2004). Furthermore, it has been demonstrated that LJM111 has anti-inflammatory activity (Grespan et al., 2012), suggesting that its biological activity is different from LJM11 and LJM17 that have now been characterized as novel neutrophil chemoattractants.”

Q. Suppl. Fig. 3: Again, there is also the experiment with LJM111. If I understand well, a relatively small number of neutrophils was used in the assay with this proteins (in contrast to other proteins tested). Could you explain this, please?

Answer: We added the same number of neutrophils for all tests. Nevertheless, in this particular experiment it looks like there were less neutrophils. Probably this was due to a loading variation, however, the number of neutrophils that was used is sufficient to test for the chemotactic activity since we can follow individual neutrophils in this assay. When there is chemotactic activity, it is very easy to observe and when there is no chemotactic activity it is also very obvious because none of the neutrophils move on a directional trajectory.

Q. Suppl. Fig. 4: Please mention what is the difference against alignments published by Sima et al. Which “new” sand fly species did you include?

Answer: We updated the yellow alignments to include three sand fly species, namely *S. schwetzi*, *N. neivai* and *P. kandelakii*. We also display the full length of the predicted mature yellow proteins for an overview of conserved areas among species, while in Sima et al, 2016 the figures display the alignment within the YRP ligand-binding sites (Fig2) or the amino acids creating tunnels in yellow related proteins (Fig S3).

We have modified the text and it now reads:

Lines 194-198, Main Text: “Alignment shows several conserved areas of amino acids across these proteins (Supplementary Fig. 4A) that were also maintained across salivary yellow proteins from other sand flies(Sima et al., 2016) (Supplementary Fig. 4B), suggesting that yellow salivary proteins from other sand flies may also exhibit neutrophil chemoattractant activity.”

Q. I was asked to comment further on previous concerns of an absent reviewer 2 and the authors' response. Below is my opinion. In comments 1 and 2, I copy first the original questions of the Reviewer 2 and my text follows in the second paragraph.

Referee 2, comment No. 1

The work is descriptive and lacks mechanistic studies. Although authors identify new chemotactic proteins in sand fly saliva, they do not determine how these proteins are inducing chemotaxis. What is the direct effect of proteins on neutrophils? Do they activate GPCR signaling, AKT activation and actin polymerization? What is the receptor on neutrophils for these proteins? Do they induce IL-8 secretion from keratinocytes in vivo? Such mechanistic insights will improve the scope of the work.

Q. Authors added experiments to test if these novel proteins are exerting their function using G protein-coupled receptors, PI3K activation and calcium influx. This is, of course, a significant improvement but I am not sure if it would be satisfactory for the reviewer. In my opinion, authors did not respond to most of his questions above. I understand that these are difficult questions but they should not be left without an answer. In my opinion, most of these questions (e.g. about the receptor on neutrophils or how these proteins are inducing chemotaxis) should be answered at least in the letter.

Answer: We have added mechanistic studies to this work which demonstrate that this is a novel chemoattractant that is working through a G-coupled receptor and activating the PI3K signaling pathway. This mechanistic insight is very significant because it places this molecule as a novel chemoattractant derived from an insect and yet it is different to other G-coupled receptor chemoattractants. Regarding the previous questions:

Q. They do not determine how these proteins are inducing chemotaxis.

Answer: The protein is inducing chemotaxis by activating neutrophils through a G-coupled receptor, resulting in signaling through the PI3K pathway and calcium influx.

Q. What is the direct effect of proteins on neutrophils?

Answer: We hypothesize that the direct effect on neutrophils is the interaction of the salivary protein with a G-coupled receptor, we do not know what part of the salivary protein is responsible for this binding nor the identity of the corresponding G-coupled receptor on neutrophils. We agree with both reviewers that it is important to obtain this information but it will require a considerable effort and time and is beyond the scope of the present study.

Q. Do they activate GPCR signaling?

Answer: Yes, the protein activates GPCR signaling

Q. AKT activation?

Answer: Yes, the protein activates the PI3K-Akt signaling pathway.

Q. actin polymerization?

Answer: It is very likely that actin polymerization is activated by this novel protein because we observed directional migration of neutrophils in response to these novel chemotactic proteins.

Q. What is the receptor on neutrophils for these proteins?

Answer: We now know that the effect on neutrophils is through interaction of the salivary protein with a G-couple receptor, we do not know the identity of the receptor.

Q. Do they induce IL-8 secretion from keratinocytes in vivo?

We do not know if these proteins induce IL8 secretion from keratinocytes in vivo, our focus was on neutrophils and the direct effect of these salivary proteins on these cells.

We thank the reviewer for posing the above-mentioned questions because the results and conclusion based on these new experiments have improved overall the quality and significance of this work.

Q. Comment No. 2.

What are the physiological levels of these proteins in sand fly saliva? How much will be injected during a sand fly bite in natural infection? Are the concentrations used for in vitro and in vivo experiments physiological or pharmacological?

Q. I appreciate detailed explanation given by the authors in their response. However, this explanation must appear also into the Results or Discussion. The paragraph added by the authors to the results section, lines 265-269, is not enough; it is very short and general. It lacks any reference and specification of sand fly species.

Answer: We thank the reviewer for this suggestion. We have added the following text to the manuscript:

Lines 273-286, Main Text: Of note, the amount of proteins in a pair of salivary glands varies by sand fly species (Cerna et al., 2002; Lestinova et al., 2017), and a sand fly delivers 70-90 % of its salivary protein content while feeding (Kato et al., 2007; Prates et al., 2008; Ribeiro, 1989). For *P. duboscqi* and *L. longipalpis*, a pair of salivary glands contains between 0.78 µg to 1 µg of protein (Cerna et al., 2002; Kato et al., 2007) and 0.5 µg to 1 µg of protein (Ferreira et al., 2016; Mondragon-Shem et al., 2020), respectively. Further, transcriptomic analysis indicates that the relative abundance of yellow proteins in the salivary glands of *L. longipalpis* and *P. duboscqi* is 16.0% (Valenzuela et al., 2004) and 7.5% (Kato et al., 2006), respectively. Based on the above, and from a recently reported estimate of the amount and proportion of yellow salivary proteins present in sand flies (Sumova et al., 2019), we estimate there is approximately 70 ng of yellow proteins in *P. duboscqi* and approximately 120 ng of yellow proteins in

L. longipalpis salivary glands. Therefore, the tested recombinant protein concentration of 0.25 μ M in an injection volume of 10 μ L (= 100 ng) is within the physiological range of yellow proteins injected during a sand fly bite (Sumova et al., 2019).

Comment No. 3.

Q. In my opinion, this comment is answered satisfactorily. However, the added figure (Supplementary Figure 4) showing the protein alignment seems to be quite similar to alignment previously published by Sima et al. (2016). Would be good to comment this and explain differences.

Answer: : We updated the sequence alignments of yellow related salivary proteins to include yellow proteins from three sand fly species: *S. schwetzi*, *N. neivai* and *P. kandelakii*. The purpose of this alignment is to display the full-length sequence of yellow salivary proteins from different sand flies providing an overview of the conserved amino acid regions among species. In Sima et al. 2016, the figures displayed the alignment showing the YRP ligand-binding sites for bioamines (Fig2) or the amino acids creating tunnels in yellow related proteins (Fig S3) but not the entire sequences of yellow related proteins. Because we do not know the yellow protein binding site for the GPCR receptor in neutrophil, we think that is appropriate to show the entire protein and the level of similarity among these yellow proteins, and to suggest that some of these yellow proteins may also exhibit chemotactic activity.

REVIEWER COMMENTS

Reviewer #3 (Remarks to the Author):

The manuscript has been revised and I am satisfied with the response to my comments.